# Global burden and future trends of head and neck cancer: a deep learning-based analysis (1980–2030)

**Qiongyuan Hu**[1☯], **Shuai Lv**[2☯], **Xinyu Wang**[1], **Peng Pan**[1], **Wei Gong**[1], **Jinyu Mei**[1*]

1 Department of Otorhinolaryngology Head and Neck Surgery, The Second Affiliated Hospital of Anhui Medical University, Hefei, Anhui, China, 2 School of Artificial Intelligence and Data Science, University of Science and Technology of China, Hefei, Anhui, China

☯ These authors contributed equally to this work.

* meijinyu@ahmu.edu.cn

**Data availability statement:** All data source files are available from the GBD database (URLs: https://www.healthdata.org/research-analysis/gbd).

**Funding:** This work was supported in study design by the Anhui Provincial Science and Technology Department under Grant 2022AH050662, and in data collection and analysis by the Anhui Provincial Postgraduate Education Quality Engineering Project under Grant 2022zyxwjxalk060, and in decision to publish by the Research Fund of Anhui

## Abstract

**Background:** Head and neck cancer (HNC) becomes a vital global health burden. Accurate assessment of the disease burden plays an essential role in setting health priorities and guiding decision-making.

**Methods:** This study explores data from the Global Burden of Disease (GBD) 2021 study, involving totally 204 countries during the period from 1980 to 2021. The analysis focuses on age-standardized incidence, mortality, and disability-adjusted life years (DALYs) for HNC. A Transformer-based model, HNCP-T, is used for the prediction of future trends from 2022 to 2030, quantified based on the estimated annual percentage change (EAPC).

**Results:** The global age-standardized incidence rate (ASIR) for HNC has escalated between 1980 and 2021, with men bearing a higher burden than women. In addition, the burden rises with age and exhibits regional disparities, with the greatest impact on low-to-middle sociodemographic index (SDI) regions. Additionally, the model predicts a continued rise in ASIR (EAPC = 0.22), while the age-standardized death rate (ASDR) is shown to decrease more sharply for women (EAPC = -0.92) than men (EAPC = –0.54). The most rapid increase in ASIR is projected for low-to-middle SDI countries, while ASDR and DALY rates are found to decrease in different degrees across regions.

**Conclusions:** The current work offers a detailed analysis of the global burden of HNC based on the GBD 2021 dataset and demonstrates the accuracy of the HNCP-T model in predicting future trends. Significant regional and gender-based differences are found, with incidence rates rising, especially among women and in low-middle SDI regions. Furthermore, the results underscore the value of deep learning models in disease burden prediction, which can outperform traditional methods.

## Introduction

In both developed and developing countries, cancer ranks as a primary reason for death, also bringing about substantial emotional, social, and economic effects [1,2]. Among them, head

Institute of translational medicine under Grant 2022zhyx-C42, and in preparation of the manuscript by the National Natural Science Foundation Incubation Program of The Second Affiliated Hospital of Anhui Medical University under Grant 2021GMFY04.

**Competing interests:** The authors do not have competing interests.

and neck cancer (HNC) is regarded as the seventh most common cancer worldwide, responsible for over 660,000 new cases and 325,000 deaths annually [3,4]. Anatomically, HNC is a group of malignant tumors occurring above the clavicle, below the skull base, and in the anterior region of the cervical vertebral column, such as thyroid cancer (TC), lip and oral cavity cancer (LOC), laryngeal cancer (LC), nasopharyngeal cancer (NPC), and other pharyngeal cancers (OPC). These cancers pose a great global health burden, with patients frequently confronting severe challenges both from the tumor itself and from treatment [5]. Head and neck tumors can invade or compress vital structures, leading to difficulty swallowing (dysphagia) when the esophagus or pharynx is affected and difficulty breathing (dyspnea) when the airway is blocked. Involvement of the larynx or vocal cords can result in speech disorders, making communication difficult. Tumor invasion of nerves may cause pain and sensory impairment, further reducing quality of life. In addition, treatments such as surgery and radiation therapy may result in functional impairment and disfigurement. These physical burdens are often accompanied by psychosocial complications such as depression and social isolation, emphasizing the severe impact of the disease on patients.

Despite the obtained advances in surgical techniques, radiotherapy [6], and molecular targeted therapies [6] that have enhanced early diagnosis and treatment, HNC is a vital public health issue, especially in regions with limited resources. Efforts made by the World Health Organization (WHO) to lower non-communicable diseases, including HNC [7], through targeted interventions addressing behavioral and dietary risk factors such as smoking, hypertension, and poor nutrition, are in consistence with the United Nations' Sustainable Development Goals to decrease premature mortality by 2030. Nevertheless, predicting future trends in HNC plays a vital role in guiding these global health initiatives [8], as current efforts to lower incidence, mortality, and disability-adjusted life years (DALYs) still remain unclear.

Reliable global predictions of HNC trends are crucial for shaping effective prevention strategies and optimizing healthcare services. Previous studies on HNC burden have been limited by small sample sizes, sociological factors [9], and regional economic disparities [10, 11], hindering a comprehensive understanding of the global impact. The GBD database provides a valuable resource for overcoming these limitations by offering standardized methodologies to assess disease burden across populations and time periods. However, to better verify and quantify the HNC burden, more comprehensive global data is required [12].

Therefore, we apply the latest GBD data from 1980 to 2021 to assess global trends in HNC. Traditional statistical methods, such as linear regression, are commonly applied in epidemiological studies but often fall short in their ability to capture the complex interactions and patterns existing in disease progression. In recent years, deep learning methods have emerged as a powerful tool in medical predictions [13]. Among these, transformer-based models have gained prominence due to their ability to capture long-range dependencies and handle complex nonlinear relationships more effectively than traditional Recurrent Neural Networks (RNNs) and Long Short-Term Memory (LSTM) networks [14,15]. This makes them particularly suitable for predicting trends in diseases such as head and neck cancers, where intricate temporal dynamics and multifaceted interactions between various risk factors are prevalent. By utilizing the advanced models, our goal is to improve the accuracy and reliability of predictions [16], ultimately contributing to the development of more feasible public health strategies.

In this study, we conduct a comprehensive analysis of global trends in HNC incidence, mortality, and DALYs from 1980 to 2021, and employ a transformer-based HNC prediction method (HNCP-T) to forecast age-standardized rates of HNC incidence, mortality, and DALYs from 2022 to 2030. By utilizing data from 1980 to 2021, we are aimed at providing

actionable insights for future decision-making, emphasizing the potential of deep learning to improve disease burden prediction and support public health policy.

## Materials and methods

As shown in Fig 1, we propose a novel strategy for predicting the global burden of HNC, referred to as HNCP-T. This section details the design and methods behind the HNCP-T strategy. In the section Data source, we introduce the used dataset. Meanwhile, the employed data preprocessing techniques are outlined in the section Data preprocessing. Further introduction of the model architecture is provided in the section Design of the predictive model, while the section Model training and validation presents the model's training process.

### Data source

In this study, we utilize the GBD 2021 dataset, providing comprehensive data on HNC incidence, mortality, DALYs, as well as age-standardized HNC incidence, mortality, and DALY rates worldwide between 1980 and 2021. In addition, the dataset also includes estimates of HNC risk factors between 2022 and 2030, stratified by age, gender, and sociodemographic index (SDI) levels.

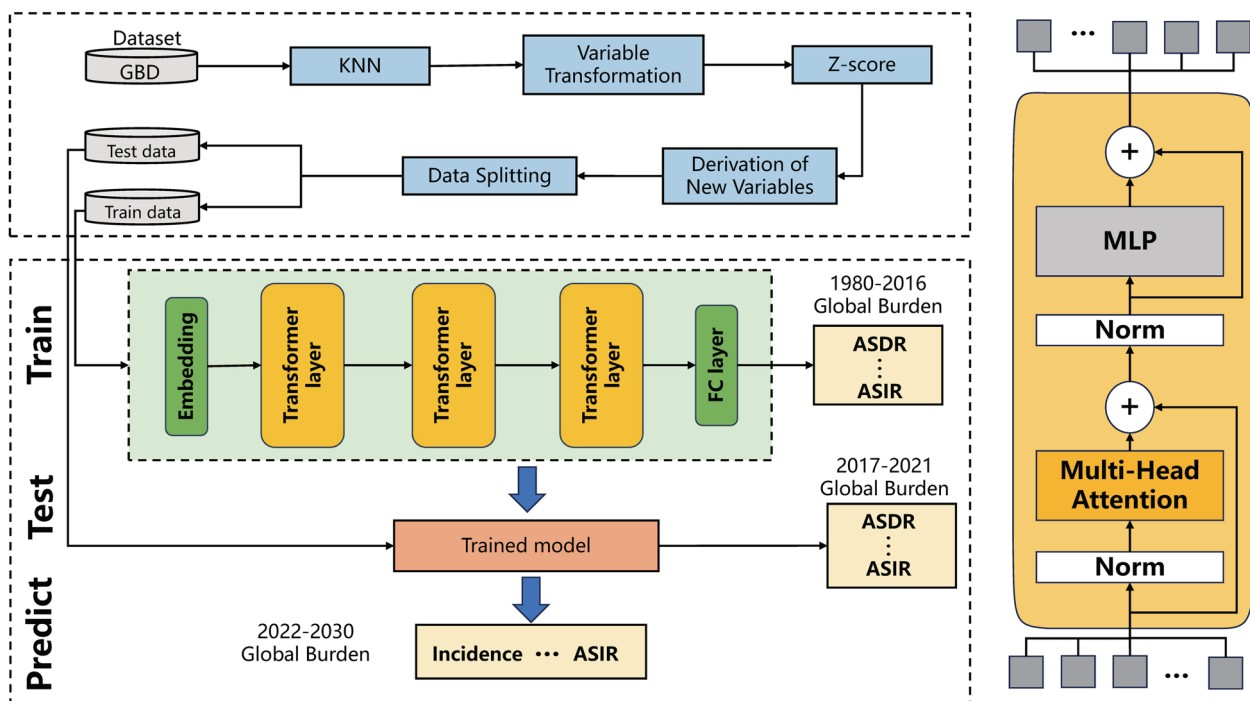

**Fig 1. Framework of the proposed method.** Data processing flow for the Global Burden of Disease (GBD) dataset. At first, the K-Nearest Neighbors (KNN) imputation technique is employed to address missing data in the GBD dataset. Then, calendar years from 1980 to 2021 are transformed into time steps for time series analysis. Subsequently, Z-score normalization is applied to standardize the dataset after deriving new variables. The dataset is split into two parts: data from 1980 to 2016 is used for training, while that from 2017 to 2021 is reserved for testing; Training, testing, and prediction strategy of the proposed model. The model is consisted of an embedding layer (yellow), transformer layers (orange, with structural details shown in the figure, and a fully connected (FC) layer (green). First, the model is trained using the training data, after which its performance is tested on the test dataset. Finally, the trained model is used for predicting the global burden of HNC from 2022 to 2030; Transformer layer architecture. Each transformer layer includes a block of layer normalization followed by multi-headed self-attention, and another block of layer normalization followed by a multi-layer perceptron (MLP). Skip connections are applied across each block to enhance model performance.

Following a comprehensive literature review and analysis of available data, the global burden of HNC data is chosen from the Global Health Data Exchange (GHDx), provided by the Institute for Health Metrics and Evaluation (IHME) [17]. Sponsored by IHME at the University of Washington, GBD is a highly exhaustive global epidemiological study on a global scale [18–20].

The specifics of the data sources are available in other references [17]. In the most recent GBD study, covering data through 2021, age standardization is conducted using the direct method, based on the global age structure of 2021. Based on on global average estimates from 1980 to 2021 obtained from the GHDx, our study further predicts age-standardized HNC incidence, death, and DALY rates for the period 2022 to 2030.

## Data preprocessing

This study performs an extensive analysis and prediction using the GBD dataset, spanning the years 1980 to 2021, with forecasts extended through 2030. As displayed in Fig 1, the data preprocessing phase involved several critical steps to sustain the integrity and quality of the data applied in the present study.

**Handling missing data with K-Nearest neighbors imputation.** To address the missing data, the K-Nearest Neighbors (KNN) imputation technique was applied. KNN imputation is a reliable method estimating missing values based on the nearest neighboring data points within the dataset, effectively preserving local structures and patterns. By maintaining these local relationships, this method enhances the predictive power of our analysis, ensuring that the underlying associations within the data are retained for more accurate predictions.

**Derivation of new variables: EAPC.** One of the variables derived in our analysis is the Estimated Annual Percentage Change (EAPC), a key metric which can be used for measuring trends in age-standardized rates (ASR) over a specified time period [21]. EAPC provides novel insight into the annual rate of change for metrics like incidence or death rates. Then, it is determined by:

$$EAPC = 100 \times (\exp(\beta) - 1) \tag{1}$$

where $\beta$ indicates the slope of the natural logarithm of the ASR, obtained from a linear regression model in which the calendar year refers to the independent variable.

Then, along with its 95% confidence interval (CI), the EAPC is applied for evaluating whether the rate is increasing or lowering over time.

**Variable selection and transformation.** Standardized indicators more effectively account for changes in population structure, offering more accurate predictions and supporting the development of improved public health strategies and policies [19,22]. By providing consistent and comparable metrics across different regions and time periods, these indicators help public health officials identify trends, allocate resources efficiently, and implement targeted interventions. Since predicting these indicators yields valuable data at both global and regional levels [18], age-standardized HNC incidence, death, and DALY rates have been selected as the primary outcome variables for this study. The predictor variables for age-standardized HNC can be detected through an extensive literature review on Google Scholar, using search terms including "risk factors", or "predictors" of HNC. Following this review, three key factors — elevated plasma glucose levels, high body mass index (BMI), and the SDI are selected for the prediction model, on the basis of data from the GHDx and the IHME. Additionally, the calendar years from 1980 to 2021 are transformed into time steps for time series analysis. Furthermore, this transformation plays an essential role in converting data

into a format suitable for time series modeling, allowing this model to effectively obtain temporal dependencies and trends.

**Standardization using Z-score.** In order to standardize the dataset, this study employs Z-score normalization, a widely applied method in data preprocessing, ensuring that all features are centered at a mean of zero with a standard deviation of one. This step is of particular importance when handling variables with varying scales, as it can prevent any single variable from disproportionately affecting the model.

The Z-score for a given data point $x_i$ is expressed by:

$$Z = \frac{x_i - \mu}{\sigma} \tag{2}$$

where $x_i$ suggests an individual data point, $\mu$ represents the mean of the dataset, and $\sigma$ denotes the standard deviation of the data .

Z-score normalization is vital in machine learning models, especially when features differ in units or scales. Through converting the data into a standardized form, Z-score normalization helps the model converge more efficiently and prevents larger-scale features from dominating the model's behavior. This standardization ensures that each feature makes equal contributions to the analysis, enhancing the overall accuracy and performance of the model.

**Data splitting.** The dataset is categorized into two parts: data from 1980 to 2015 is applied as the training set, while data from 2016 to 2021 is reserved as the test set. When compared with shuffling and randomly splitting the dataset, this approach ensures that the model is trained on historical data and evaluated on more recent data, which allows for a more accurate assessment of its predictive performance on unseen data, referring to the test dataset that the model has not encountered during training.

Through carefully following these preprocessing steps, it can be ensured that the dataset is optimized for analysis and predictive modeling, offering a solid basis for reliable and valid results.

## Design of the predictive model

The architecture of our predictive model, as displayed in Fig 1, consists of several key components specifically designed to enhance the handling of temporal and contextual dependencies within the dataset. Unlike traditional RNNs and LSTMs, which process data sequentially and may struggle with long-term dependencies, the transformer-based architecture employs self-attention mechanisms that allow for parallel processing and more efficient capturing of long-range interactions [15]. This architectural choice not only improves computational efficiency but also enhances the model's ability to discern complex patterns within the data.

- **Embedding layer:** This initial layer converts categorical input data into dense vector representations. Reducing dimensionality and noise is crucial in this predictive context because high-dimensional data, especially with categorical variables, can lead to overfitting and obscure important patterns. By embedding categorical inputs, the model simplifies the data representation, removing irrelevant variations and highlighting essential relationships. This reduction in complexity not only improves the model's predictive accuracy but also enhances its ability to generalize to new data, offering a clearer, more reliable understanding of the underlying patterns in HNC trends. This dense representation offers a richer set of features for the subsequent layers, therefore enhancing the model's capability of effectively processing and learning from the input data.

- **Transformer layers:** Following the Embedding layer, the model employs three Transformer layers, as displayed in Fig 1. These layers are vital for obtaining complex dependencies and interactions across different time steps in the input data. The multi-head self-attention mechanism within the Transformer layers allows the model to concentrate on multiple segments of the input sequence simultaneously, enabling it to recognize vital features regardless of their positional context. Moreover, this capability is vital for modeling the intricate dynamics present in time series data. This ensures that the model can accurately obtain long-range dependencies.
- **Fully connected layer:** Concluding the architecture is a Fully Connected (FC) layer that can aggregate the high-dimensional features extracted by the Transformer layers. This layer combines information by taking the rich, multi-dimensional representations learned through the Transformer's self-attention mechanism and integrating them into a more compact form. It does so by weighting and summing these features, effectively capturing the complex feature interactions across the entire sequence. Then, the FC layer maps these aggregated features to the desired output dimensions, transforming the complex relationships into predictions tailored to the specific forecasting task. This process ensures that the final predictions leverage all the relevant information captured by the preceding Transformer layers.

The sequential configuration of these components—from the initial data embedding to processing through Transformer layers, and finally, aggregation in the FC layer—is designed to maximize the extraction and interpretation of useful patterns for time series forecasting. This architecture can ensure that each stage of the model contributes to a comprehensive understanding and processing of the input data, culminating in accurate and reliable predictions. As presented in Fig 1, the prediction model consists of an Embedding layer, followed by three Transformer layers, and a final FC layer. Furthermore, this architecture uses the power of Transformer layers to effectively handle sequential data, ensuring that the model captures both temporal dynamics and contextual relationships, which are vital for obtaining high prediction accuracy in time series forecasting tasks.

In summary, the methodological approach employed in this study, which incorporates deep learning methods, plays an important role in capturing the complex temporal patterns and relationships within the HNC data. By utilizing age-standardized incidence, death, and DALY rates as key outcome variables, this approach allows for a more accurate and comprehensive analysis of the global burden of HNC. These methods not only improve the reliability of predictions but also provide valuable insights that can inform public health strategies and policy development aimed at mitigating the impact of HNC globally.

## Model training and validation

To explore and predict the global burden of HNC from 1980 to 2030 using the GBD dataset, this study implements a structured training strategy consisting of multiple stages including data preparation, model training, and testing.

Our model is trained on data from 1980 to 2015, including comprehensive information on global HNC incidence, mortality, DALYs, ASIR, ASDR, and age-standardized DALYs. This time range is chosen to ensure that the model could capture long-term trends and patterns in the burden of HNC across regions and demographics.

The model is trained with the Mean Absolute Error (MAE) as the loss function, delimited by the following formula:

$$\text{MAE} = \frac{1}{n} \sum_{i=1}^{n} |y_i - \hat{y}_i| \tag{3}$$

where $y_i$ represents the actual observed value, $\hat{y}_i$ refers to the predicted value, and $n$ suggests the number of observations. MAE is chosen due to its robustness in the presence of outliers and its straightforward interpretation as the average magnitude of errors in predictions. This loss function is particularly suitable for our task, as it can effectively handle outliers and provide a robust measure of prediction accuracy.

After training the model, we tested its performance using the obtained data from 2016 to 2021. During this period, this test dataset is fed into the trained model to generate predictions for global HNC burden metrics. Specifically, the model predicted global incidence, mortality, DALYs, ASIR, ASDR, as well as age-standardized DALYs. This testing phase is vital for demonstrating the model's accuracy and generalization, ensuring its ability to reliably predict the global burden of HNC on unseen data.

The model's performance on the test set provided essential insights into its generalization ability and allowed us to identify areas where the model could be improved. The evaluation on the test set guided the training process in several key ways: Firstly, the results from the test set reveal specific areas where the model's predictions are less accurate. Also, by analyzing the test set predictions, we identify which predictor variables (such as plasma glucose levels, BMI, and SDI) have the most significant impact on the model's accuracy. Furthermore, based on the performance discrepancies observed on the test set, we employ regularization techniques, such as dropout and L2 regularization, to prevent overfitting. This ensured that the model maintained its predictive power even on unseen data, improving generalization. Last but not least, the test set results help us identify specific cases of large prediction errors. By focusing on these outliers, we could modify the training process to give more attention to the problematic time frames or regions, thereby refining the model's accuracy.

Based on a validated and trained model, we proceeded to predict the global burden of HNC from 2022 to 2030. The predictions generated by the model for this period provide novel insights into potential future trends in HNC incidence, mortality, and DALYs, as well as their age-standardized rates. These predictions are crucial for global health planning and policy-making, offering a data-driven foundation for anticipating and addressing the future burden of HNC.

Moreover, the structured approach to model training and testing ensures that the model not only obtains historical trends accurately but also offers reliable forecasts for future global health outcomes which are associated with HNC.

## Results

This section analyzes the global burden of HNC and predicts its future trends. We examine global incidence, mortality, DALYs, ASIR, ASDR, and age-standardized DALYs of HNC on the basis of the GBD dataset, and the analysis results are shown in Section. In order to offer a clearer understanding of future global HNC burden, this study forecasts ASIR, ASDR, and age-standardized DALYs from 2022 to 2030 presented in Section.

### Analyze the global trend of burden of HNC

Tables 1 and 2 and Figs 2–5 present the incidence, mortality, DALYs, ASIR, ASDR, age-standardized DALYs, and EAPC of HNC worldwide from 1980 to 2021 categorized by gender, SDI and age.

**Table 1. Trends in global head and neck cancer (HNC) burden from 1980 to 2021 by sex and 5 sociodemographic index (SDI) regions, along with the overall trend.**

| Characteristics | 1980 | | 2021 | | | 1990 | | 2021 | | | 1980 | | 2021 | | |
|---|---|---|---|---|---|---|---|---|---|---|---|---|---|---|---|
| | Incidence cases | ASIR | Incidence cases | ASIR | EAPC | Death cases | ASDR | Death cases | ASDR | EAPC | DALYs cases | Age-standardized DALY rate | DALYs cases | Age-standardized DALY rate | EAPC |
| Global | 529923 (500609, 562749) | 12.69 (12.00, 13.46) | 1160696 (1064793, 1256106) | 13.40 (12.30, 14.51) | 0.12 (0.06, 0.19) | 250174 (228321, 274579) | 7.74 (7.08, 8.46) | 544223 (499649, 587847) | 6.30 (5.78, 6.80) | −0.65 (−0.72, −0.57) | 9793094 (9102698, 10529251) | 228.30 (212.49, 245.34) | 15597835 (14180450, 16948531) | 179.37 (162.94, 194.93) | −0.94 (−1.04, −0.84) |
| **Sex** | - | | | | | | | | | | | | | | |
| Female | 172514 (156577, 189403) | 7.82 (7.11, 8.57) | 410139 (364576, 467404) | 9.20 (8.18, 10.50) | 0.50 (0.45, 0.55) | 70025 (59595, 80400) | 4.07 (3.48, 4.66) | 154535 (136440, 175856) | 3.38 (2.99, 3.85) | −0.56 (−0.63, −0.5) | 2594108 (2260147, 2926996) | 116.08 (101.31, 130.87) | 4306733 (3800148, 4926813) | 96.70 (85.19, 110.87) | −0.75 (−0.87, −0.64) |
| Male | 357409 (334798, 382395) | 18.26 (17.12, 19.50) | 750556 (678497, 821397) | 18.11 (16.39, 19.80) | −0.09 (−0.17, −0.01) | 180149 (162855, 201210) | 12.17 (11.02, 13.51) | 389688 (350279, 426561) | 9.58 (8.64, 10.47) | −0.73 (−0.81, −0.66) | 7198985 (6605314, 7843279) | 349.61 (321.37, 380.23) | 11291102 (10040997, 12430877) | 268.06 (238.61, 294.81) | −1.01 (−1.11, −0.92) |
| **SDI** | - | | | | | | | | | | | | | | |
| High SDI | 165785 (160094-, 170722) | 15.70 (15.18, 16.17) | 291197 (273050, 305275) | 15.96 (15.10, 16.73) | 0.13 (0.03, 0.22) | 55256 (52908, 57528) | 5.99 (5.72, 6.23) | 82228 (75416, 86982) | 4.00 (3.71, 4.21) | −1.08 (−1.12, −1.03) | 1705796 (1644835, 1766881) | 163.19 (157.42, 169.06) | 1984006 (1863100, 2094436) | 107.97 (101.78, 113.89) | −1.39 (−1.43, −1.34) |
| High-middle SDI | 138677 (129740, 147834) | 13.46 (12.59, 14.35) | 253047 (225779, 283680) | 13.38 (11.92, 15.05) | −0.15 (−0.26, −0.04) | 61768 (56204, 67802) | 7.51 (6.85, 8.21) | 99787 (89598, 110856) | 5.08 (4.57, 5.65) | −1.25 (−1.43, −1.06) | 2527577 (2340654, 2724479) | 242.70 (224.86, 261.38) | 2772936 (2479836, 3098944) | 144.77 (129.46, 161.56) | −2.01 (−2.14, −1.88) |
| Middle SDI | 117058 (106789, 128510) | 10.23 (9.36, 11.22) | 325787 (286610, 363718) | 11.76 (10.35, 13.12) | 0.36 (0.21, 0.51) | 64013 (56018, 72824) | 8.10 (7.15, 9.15) | 165260 (147511, 183181) | 6.13 (5.49, 6.79) | −0.85 (−0.93, −0.77) | 2764156 (2520521, 3022844) | 228.68 (209.01, 249.96) | 4765368 (4234644, 5289409) | 169.38 (150.72, 187.78) | −1.15 (−1.27, −1.02) |
| Low-middle SDI | 82831 (72080, 95567) | 12.30 (10.73, 14.15) | 227393 (199384, 256535) | 14.71 (12.95, 16.53) | 0.55 (0.47, 0.64) | 52640 (44095, 62547) | 10.58 (8.83, 12.53) | 153701 (136169, 171760) | 10.47 (9.30, 11.67) | −0.01 (−0.04, 0.02) | 2125437 (1845797, 2451250) | 297.34 (257.85, 342.72) | 4691848 (4098665, 5277231) | 294.46 (258.92, 329.92) | −0.05 (−0.1, 0) |
| Low SDI | 24971 (20526, 29895) | 9.84 (8.13, 11.74) | 62243 (51675, 75164) | 10.70 (9.00, 12.70) | 0.15 (0.04, 0.26) | 16237 (13024, 19849) | 8.45 (6.82, 10.29) | 42733 (35841, 50521) | 8.15 (6.90, 9.55) | −0.14 (−0.19, −0.09) | 659382 (542490, 786597) | 243.05 (200.14, 290.57) | 1369702 (1135538, 1639294) | 224.25 (187.66, 266.01) | −0.39 (−0.47, −0.31) |

**Table 2. Trends in global head and neck cancer (HNC) burden from 1980 to 2021 by 19 age groups (5-year intervals)**

| Characteristics | Incidence cases | | | Death cases | | | DALYs cases | | |
|---|---|---|---|---|---|---|---|---|---|
| | 1980 | 2021 | EAPC | 1990 | 2021 | EAPC | 1980 | 2021 | EAPC |
| **Age** | | | | | | | | | |
| 5–9 years old | 1047 (891, 1213) | 917 (724, 1149) | −0.51 (−0.81, −0.21) | 387 (299, 498) | 193 (145, 246) | −1.94 (−2.19, −1.69) | 33669 (28051, 39817) | 16426 (12361, 20980) | −2.54 (−2.81, −2.27) |
| 10–14 years old | 1689 (1472, 1885) | 2107 (1718, 2481) | 0.44 (0.3, 0.58) | 688 (555, 829) | 506 (409, 609) | −0.87 (−1.02, −0.72) | 53879 (46523, 61426) | 40241 (32490, 48580) | −1.23 (−1.4, −1.06) |
| 15–19 years old | 3755 (3373, 4287) | 5333 (4311, 6768) | 1.07 (0.96, 1.17) | 1302 (1131, 1513) | 1305 (1043, 1615) | −0.33 (−0.45, −0.22) | 111297 (99860, 125697) | 97179 (77823, 120399) | −0.58 (−0.66, −0.5) |
| 20–24 years old | 5547 (4892, 6317) | 9573 (7840, 12177) | 1.95 (1.81, 2.09) | 1616 (1394, 1858) | 2320 (1855, 2874) | 0.64 (0.47, 0.82) | 153067 (134041, 173274) | 161573 (129187, 200770) | 0.21 (0.15, 0.26) |
| 25–29 years old | 8243 (7279, 9417) | 16131 (13501, 19512) | 2.35 (2.21, 2.49) | 2397 (2044, 2780) | 3377 (2782, 4012) | 0.76 (0.62, 0.9) | 187653 (163202, 211881) | 219580 (180781, 261012) | 0.42 (0.28, 0.55) |
| 30–34 years old | 13104 (11774, 14575) | 29883 (25295, 35154) | 2.54 (2.42, 2.65) | 3407 (2940, 3874) | 5953 (5051, 6850) | 0.91 (0.73, 1.09) | 270405 (239591, 301198) | 358425 (303154, 413488) | 0.58 (0.4, 0.77) |
| 35–39 years old | 22984 (20792, 25234) | 45628 (39751, 52007) | 2.11 (2.03, 2.18) | 6011 (5194, 6770) | 10423 (9028, 11806) | 1.02 (0.8, 1.25) | 474439 (420234, 528162) | 572013 (494026, 647561) | 0.39 (0.28, 0.51) |
| 40–44 years old | 33010 (30471, 35813) | 65252 (57412, 73253) | 2.02 (1.89, 2.14) | 12245 (10874, 13785) | 19955 (17574, 22289) | 1.12 (0.94, 1.3) | 732442 (665456, 803184) | 985303 (865503, 1101739) | 0.58 (0.47, 0.69) |
| 45–49 years old | 43786 (40985, 47055) | 90374 (80591, 100235) | 2.34 (2.16, 2.51) | 20271 (18216, 22747) | 32445 (28630, 36225) | 1.24 (1.15, 1.33) | 973493 (897265, 1062712) | 1435528 (1265404, 1600305) | 1.13 (0.99, 1.27) |
| 50–54 years old | 64014 (59998, 68525) | 126867 (114657, 139649) | 2.43 (2.32, 2.53) | 30685 (27514, 34349) | 51359 (45971, 56744) | 1.21 (1.15, 1.27) | 1384495 (1275024, 1498394) | 2019271 (1804552, 2232927) | 1.30 (1.21, 1.4) |
| 55–59 years old | 75925 (71211, 81242) | 156563 (143801, 170099) | 2.55 (2.34, 2.77) | 35450 (31962, 39939) | 68386 (62235, 74689) | 1.36 (1.21, 1.5) | 1512491 (1399152, 1641559) | 2361425 (2145604, 2582476) | 1.50 (1.28, 1.73) |
| 60–64 years old | 78668 (74242, 83865) | 156165 (143609, 168614) | 2.41 (2.11, 2.71) | 34141 (30786, 38028) | 75425 (68754, 82057) | 1.61 (1.43, 1.8) | 1426611 (1324404, 1540811) | 2246001 (2045810, 2443922) | 1.49 (1.2, 1.79) |
| 65–69 years old | 66935 (62831, 71132) | 152609 (140500, 165205) | 2.41 (2.14, 2.69) | 34910 (31803, 38355) | 78282 (71354, 85159) | 1.81 (1.63, 1.98) | 1068276 (988350, 1152554) | 1968358 (1794266, 2141499) | 1.60 (1.32, 1.87) |
| 70–74 years old | 46960 (43972, 50017) | 124579 (114783, 134690) | 2.66 (2.49, 2.83) | 29360 (26881, 31957) | 68882 (62955, 75015) | 1.92 (1.81, 2.03) | 674351 (623753, 728083) | 1430430 (1306394, 1555741) | 1.91 (1.73, 2.09) |
| 75–79 years old | 34256 (32110, 36242) | 81402 (74197, 87900) | 3.11 (3.02, 3.19) | 19836 (18248, 21431) | 49908 (45526, 54155) | 2.29 (2.23, 2.34) | 422616 (393482, 451954) | 830096 (756148, 901922) | 2.43 (2.36, 2.5) |
| 80–84 years old | 18823 (17055, 20136) | 51963 (45601, 56140) | 3.58 (3.4, 3.76) | 10660 (9743, 11576) | 37425 (33243, 40562) | 3.00 (2.89, 3.11) | 207827 (189420, 224313) | 485802 (431621, 526706) | 2.98 (2.81, 3.15) |
| 85–89 years old | 8446 (7434, 9110) | 30510 (25437, 33464) | 4.32 (4.11, 4.54) | 4861 (4351, 5322) | 23515 (20016, 25726) | 3.90 (3.76, 4.04) | 79480 (70876, 86168) | 242757 (206719, 265180) | 3.69 (3.48, 3.91) |
| 90–94 years old | 2296 (1932, 2511) | 11884 (9484, 13201) | 5.34 (5.13, 5.55) | 1569 (1343, 1739) | 10924 (8818, 12083) | 4.93 (4.79, 5.06) | 21756 (18526, 23818) | 97528 (79063, 107940) | 4.78 (4.57, 5) |
| 95+ years old | 434 (338, 489) | 2954 (2165, 3389) | 6.57 (6.5, 6.65) | 376 (298, 440) | 3639 (2663, 4165) | 5.99 (5.89, 6.1) | 4846 (3821, 5470) | 29900 (21978, 34190) | 6.20 (6.13, 6.27) |

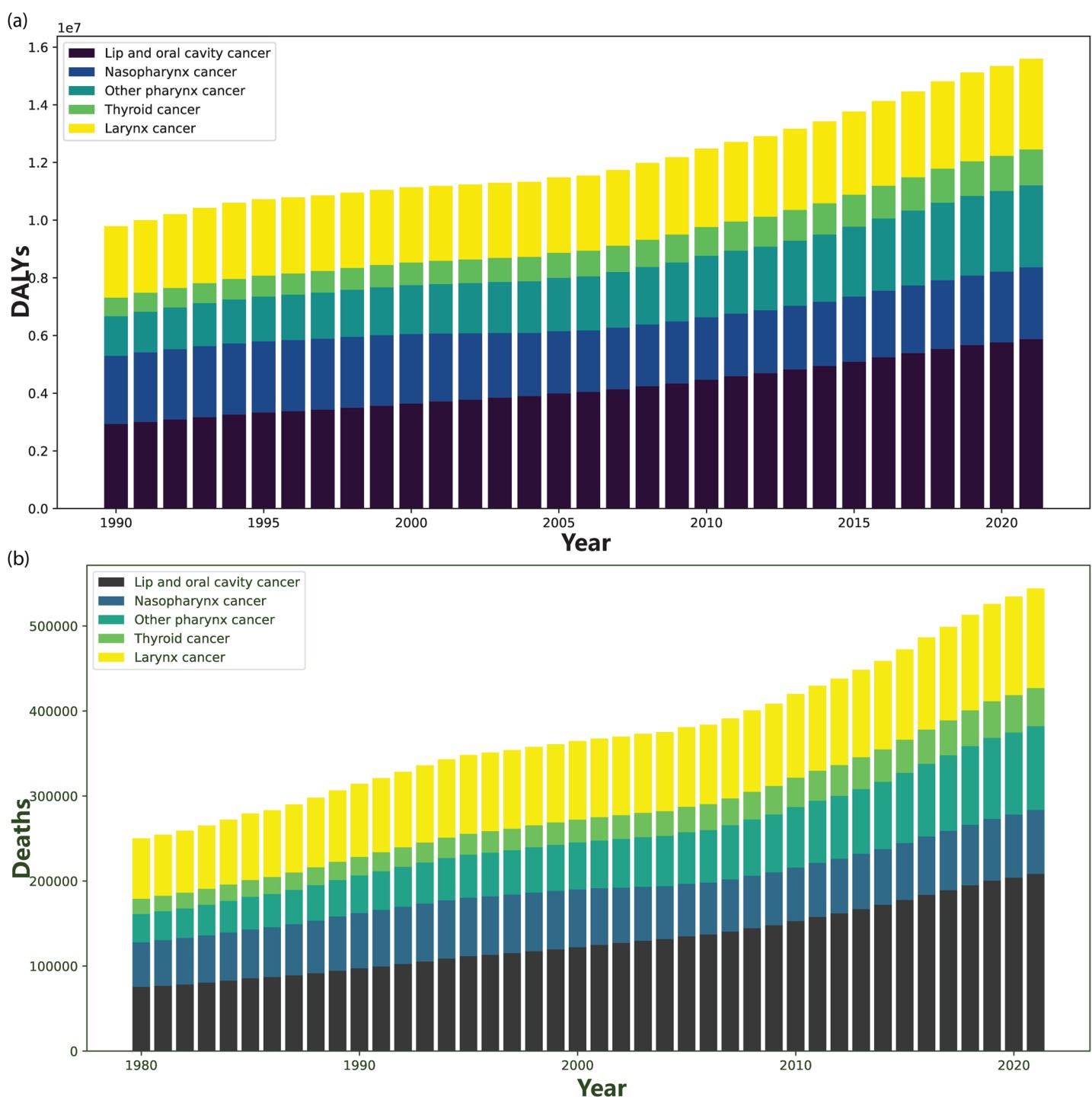

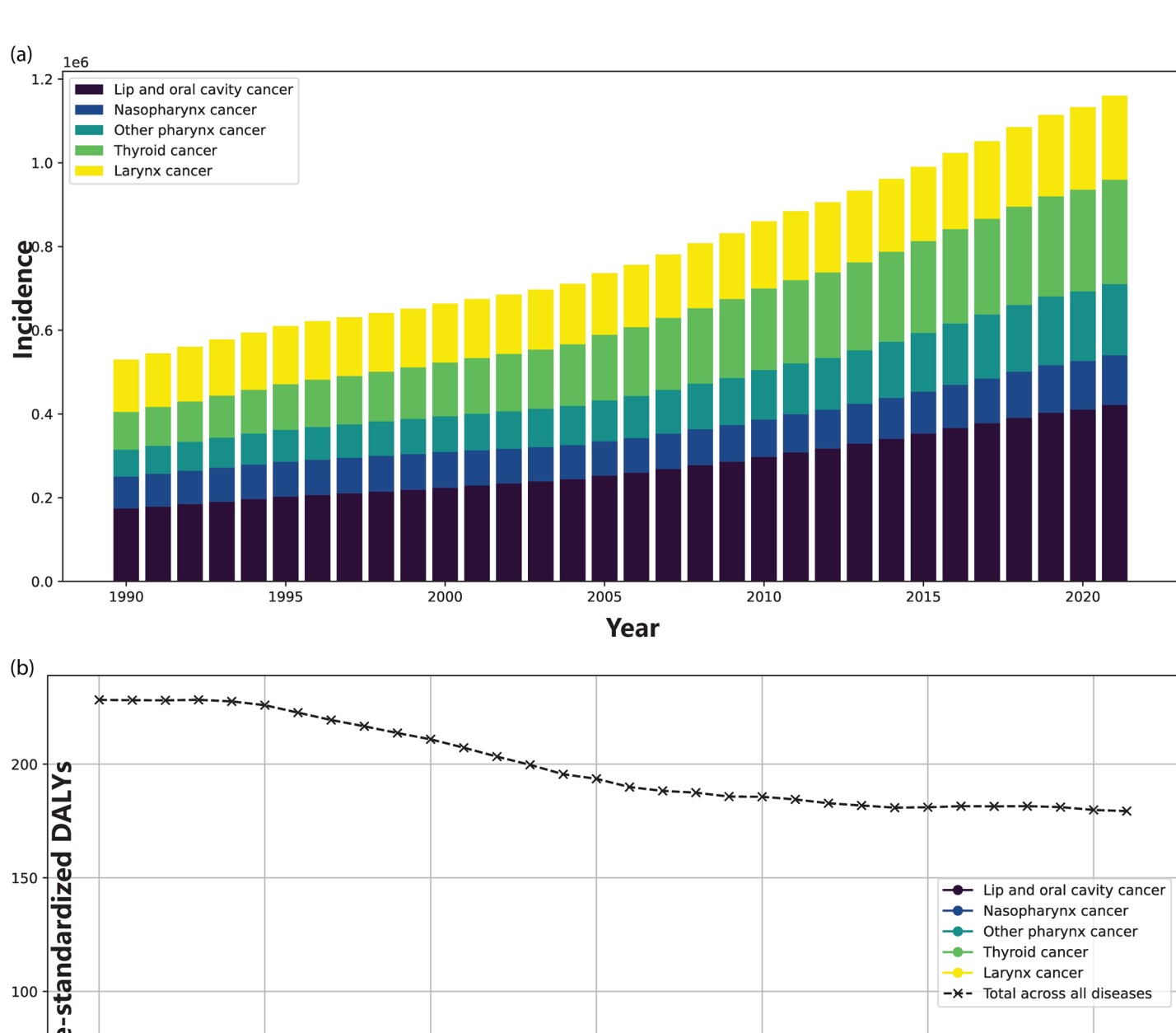

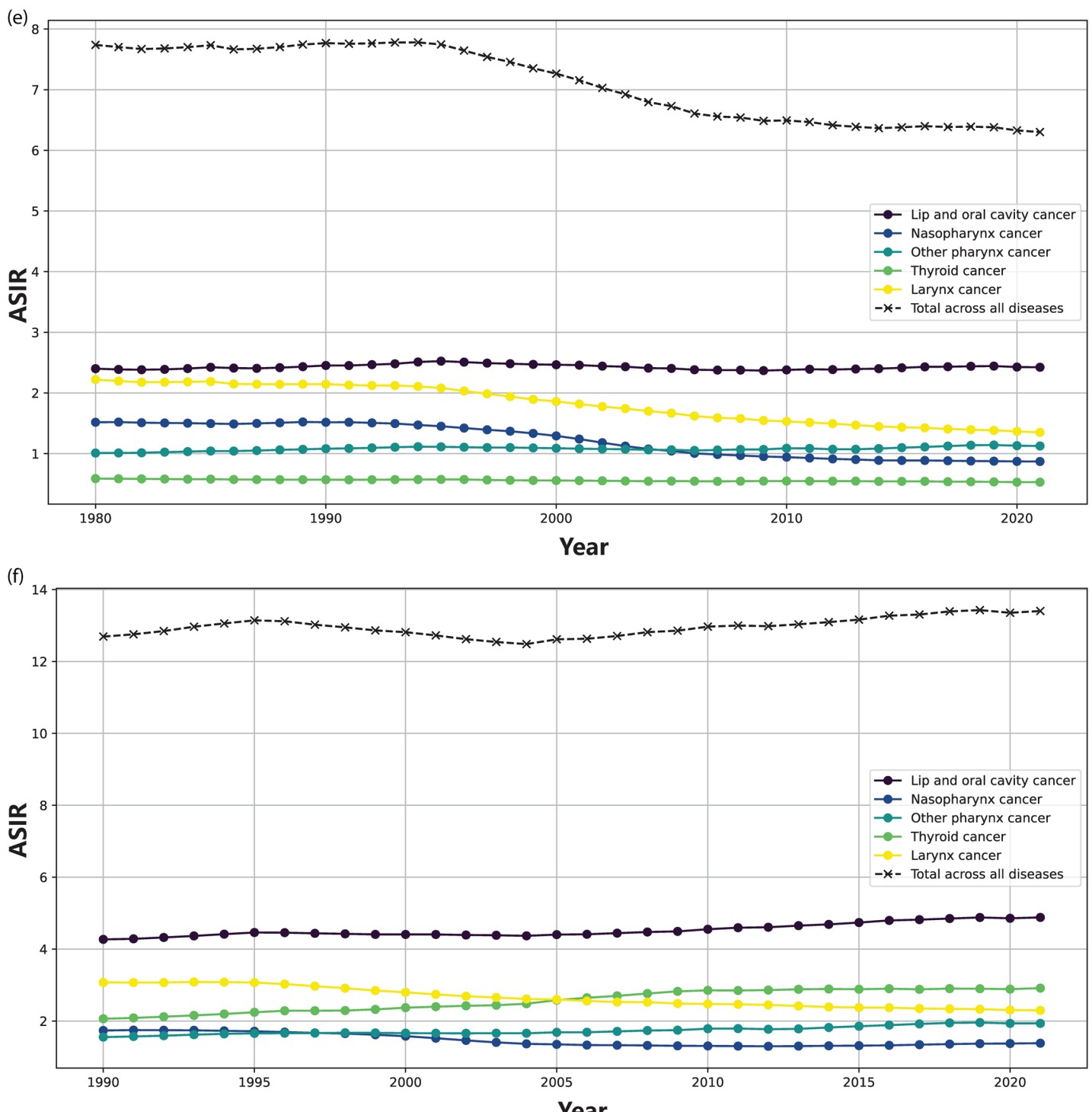

**Fig 2. Trends in the global disease burden of head and neck cancer (HNC) from 1980 to 2021.** (a) The trends in disability-adjusted life years (DALYs) for different subtypes of HNC, and the overall trend in DALYs for HNC. (b) The trends in Deaths for different subtypes of HNC, and the overall trend in Deaths for HNC. (c) The trends in incidence for different subtypes of HNC, and the overall trend in incidence for HNC. (d) The trends in age-standardized DALYs for different subtypes of HNC, and the overall trend in age-standardized DALYs for HNC. (e) The trend in age-standardized death rate (ASDR) for different subtypes of HNC, and the overall trend in ASDR for HNC. (f) The trend in age-standardized incidence rate (ASIR) for different subtypes of HNC, and the overall trend in ASIR for HNC.

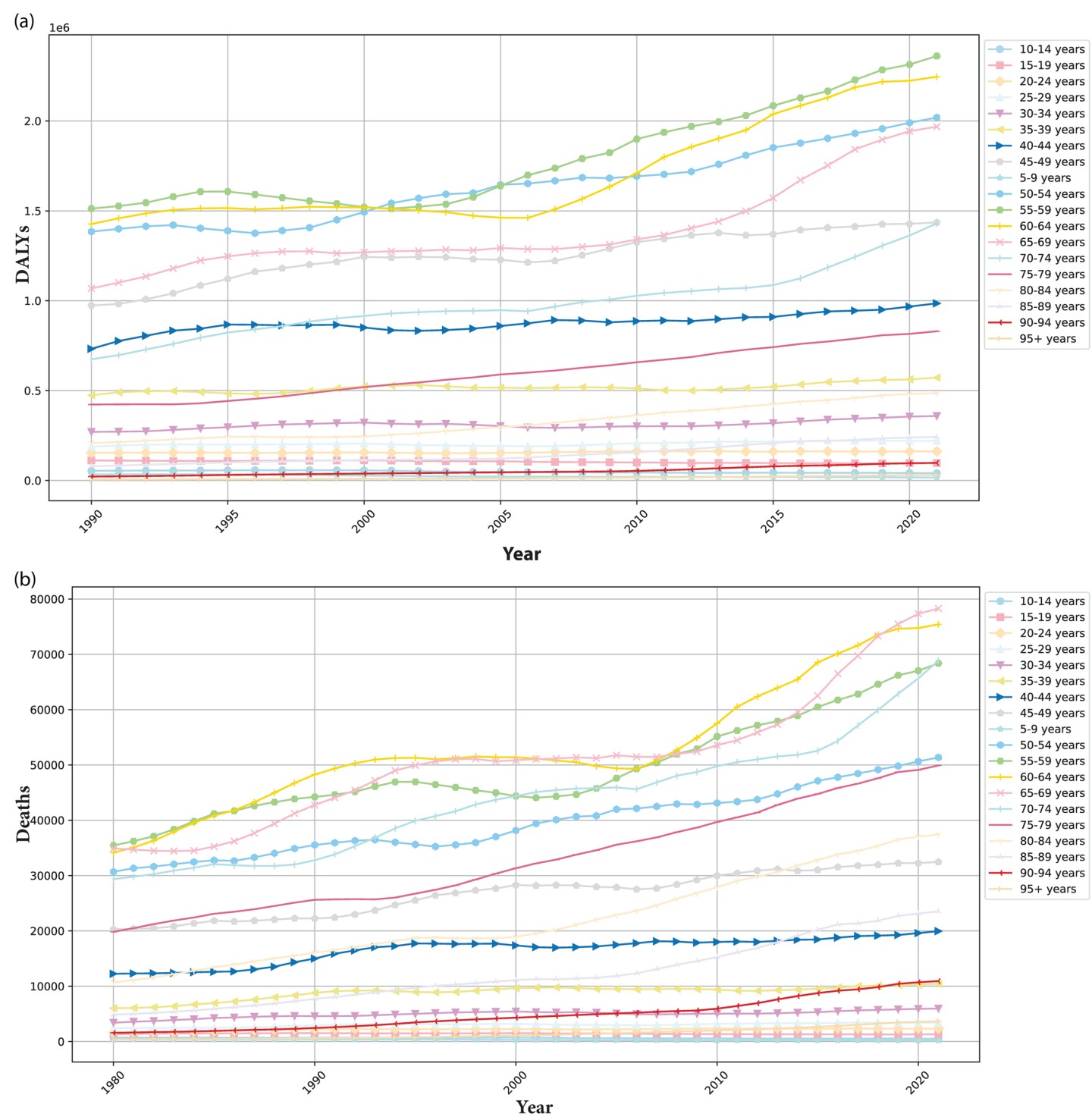

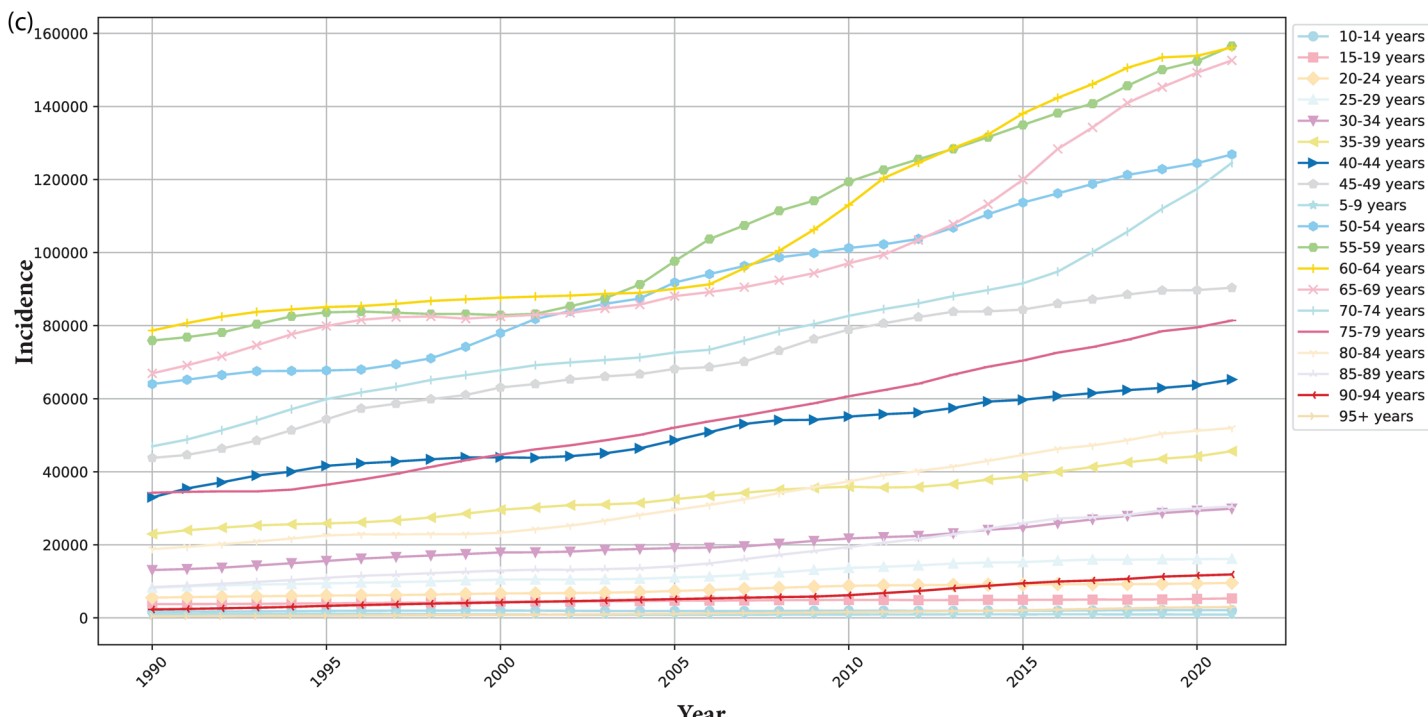

**Fig 3. Trends in the global disease burden of head and neck cancer (HNC) from 1980 to 2021, by ages.** (a) The trend in disability-adjusted life years (DALYs) of HNC across different age groups (5-year intervals). (b) The trend in Deaths of HNC across different age groups (5-year intervals). (c) The trend in incidence of HNC across different age groups (5-year intervals).

**The global trend of burden of HNC.** In 2021, the global number of DALYs of HNC is 15,597,835 (95% CI: 14,180,450–16,948,531 ), and age-standardized DALYs is 179.37 (95% CI: 162.94–194.93) (Table 1, Fig 2a). The number of deaths is 544, 223 (95% CI: 499,649–587,847), causing an ASDR of 6.30 (95% CI: 5.78–6.80) (Table 1, Fig 2b). The number of incidence is 1, 160, 696 (95% CI: 1,064,793–1,256,106), with an ASIR 13.40 (95% CI: 12.30–14.51) (Table 1, Fig 2c). By analyzing the age-standardized trends, it is evident that the ASIR has exhibited a progressive rise, with EAPC of 0.12 (95% CI: 0.06–0.19) (Table 1, Fig 2f). By contrast, the ASDR and age-standardized DALYs have revealed a declining trend, with EAPC of −0.65 (95% CI: [−0.72]–[−0.57]) and −0.94 (95% CI: [−1.04]–[−0.84]), respectively (Table 1, Fig 2d and 2e).

**The global trend of burden of HNC by SDI.** AS is shown in Table 1. In 2021, compared with 1980, there was a general decrease in ASDR and age-standardized DALYs across all SDI regions, while incidence, death, DALYs, and ASIR exhibited an increasing trend. Notably, in 2021, the high SDI level region exhibited the highest ASIR for HNC, with a value of 15.96 (95% CI: 15.10 – 16.73) (Table 1 and Fig 5f). However, it also displayed the lowest ASDR 4.00 (95% CI: 3.71 – 4.21) and the lowest age-standardized DALYs 107.97 (95% CI: 101.78–113.89) (Table 1 and Fig 5). According to the data from 1980 to 2021, a comprehensive analysis of different SDI regional data reveals a slowly increasing trend in the incidence, mortality, and DALYs (Fig 5a–5c). Conversely, there is a decreasing trend in ASDR and age-standardized DALYs, except in the low-medium or low SDI regions with stable trend (Fig 5d and 5e).

(a)

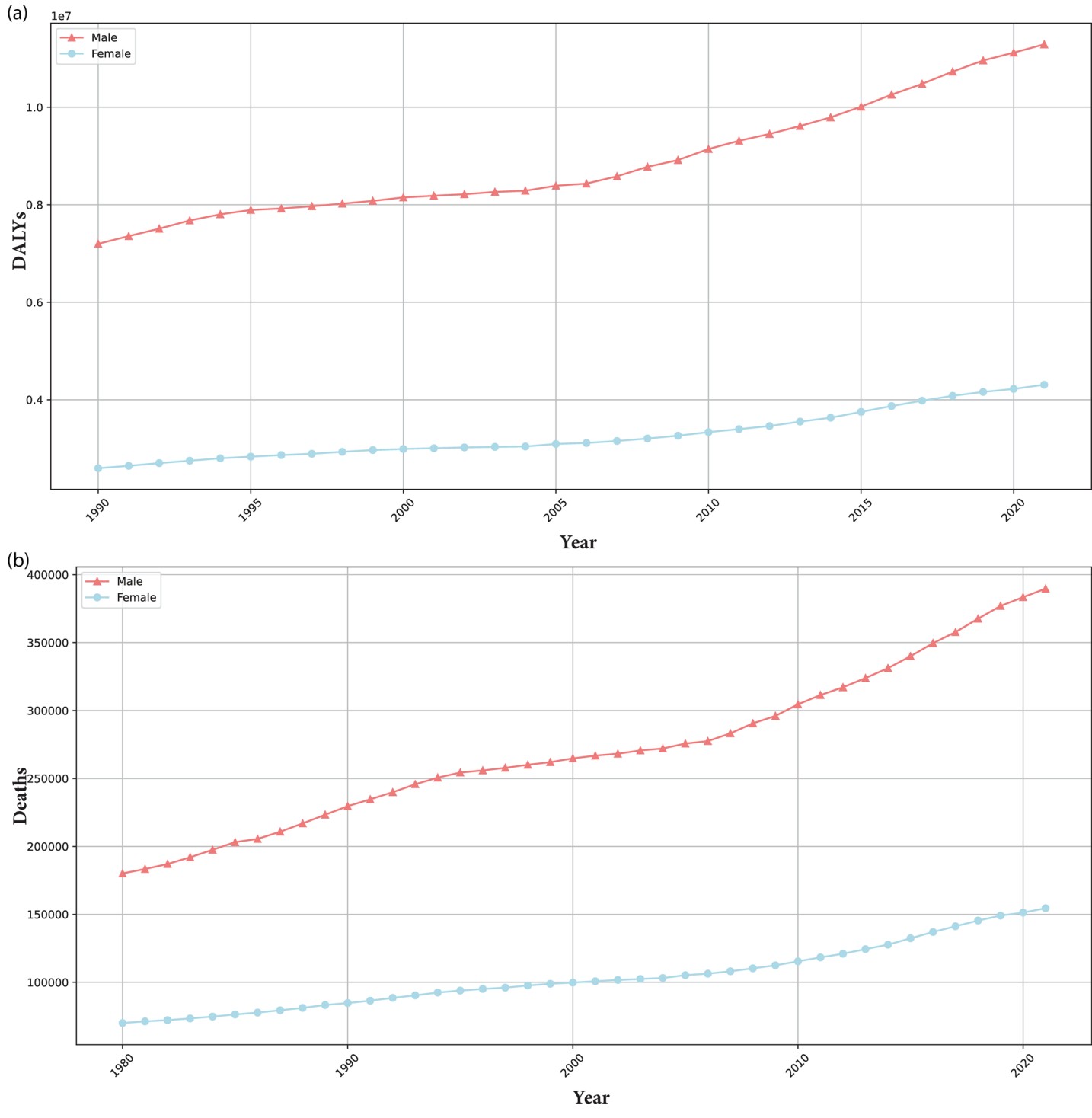

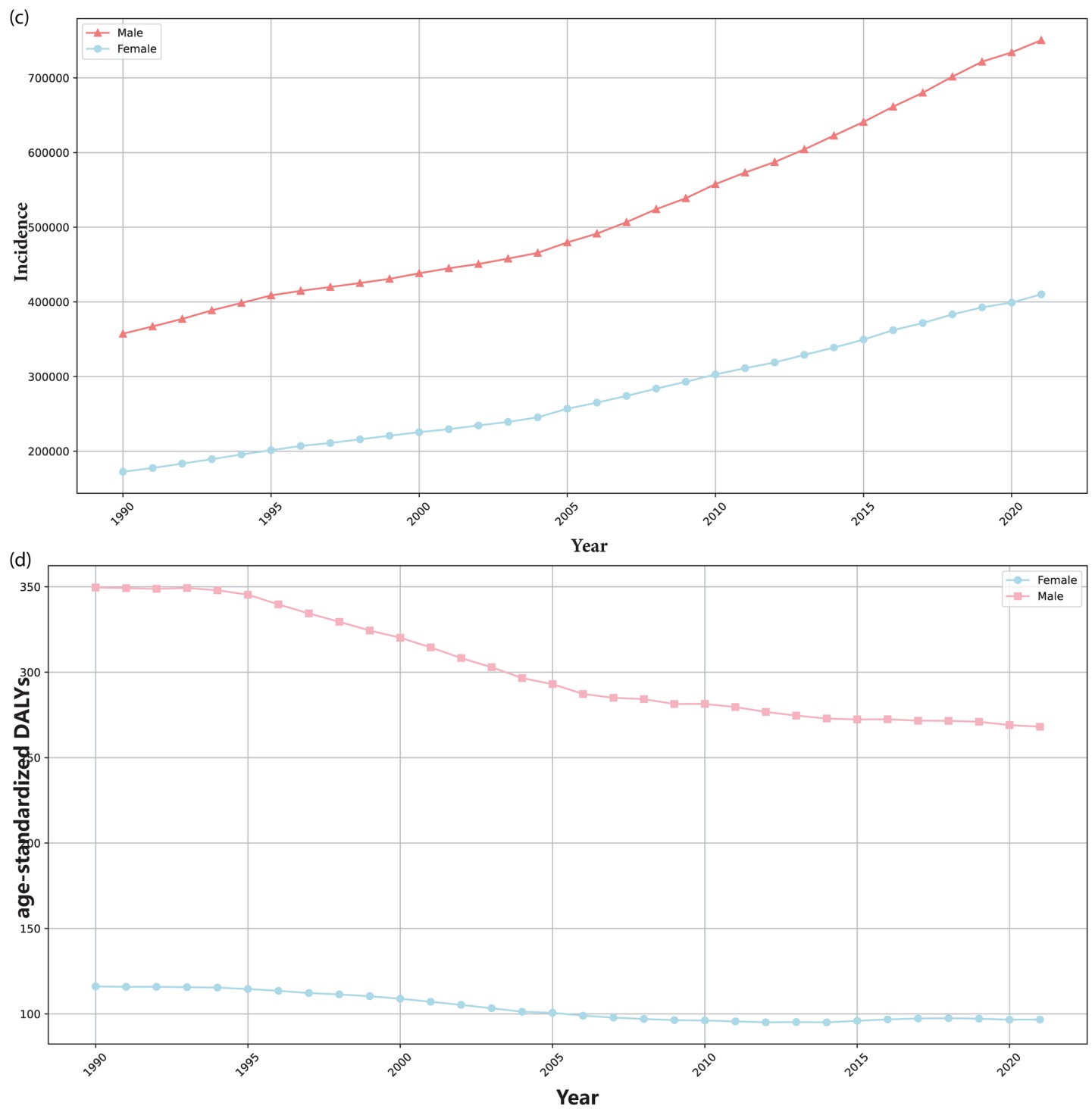

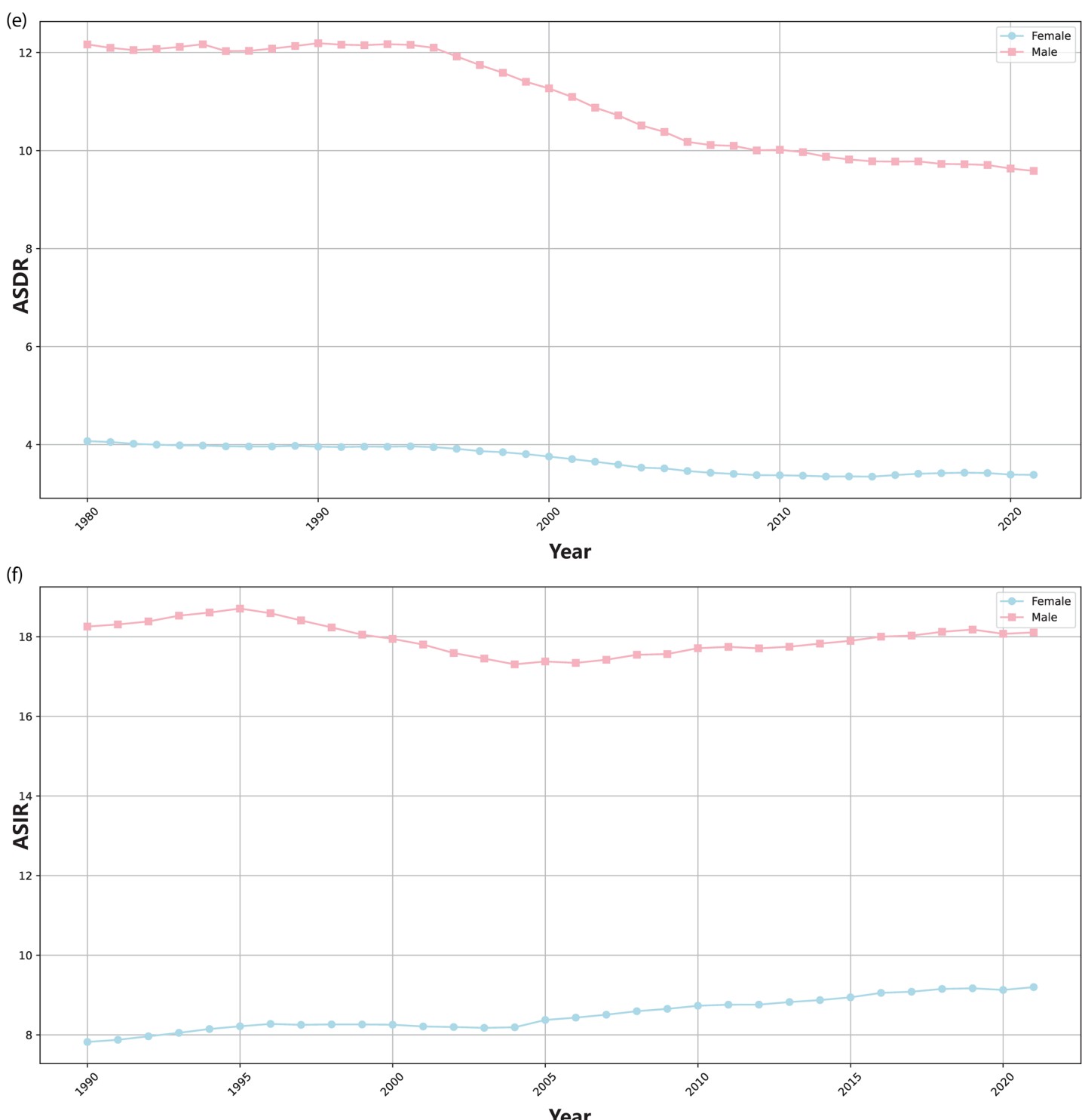

**Fig 4. Trends in the global disease burden of head and neck cancer (HNC) from 1980 to 2021, by genders.** (a) The trend in disability-adjusted life years (DALYs) of HNC across different gender groups (male and female). (b) The trend in Deaths of HNC across different gender groups (male and female). (c) The trend in incidence of HNC across different gender groups (male and female). (d) The trend in age-standardized DALYs of HNC across different gender groups (male and female). (e) The trend in age-standardized death rate (ASDR) of HNC across different gender groups (male and female). (f) The trend in age-standardized incidence rate (ASIR) of HNC across different gender groups (male and female).

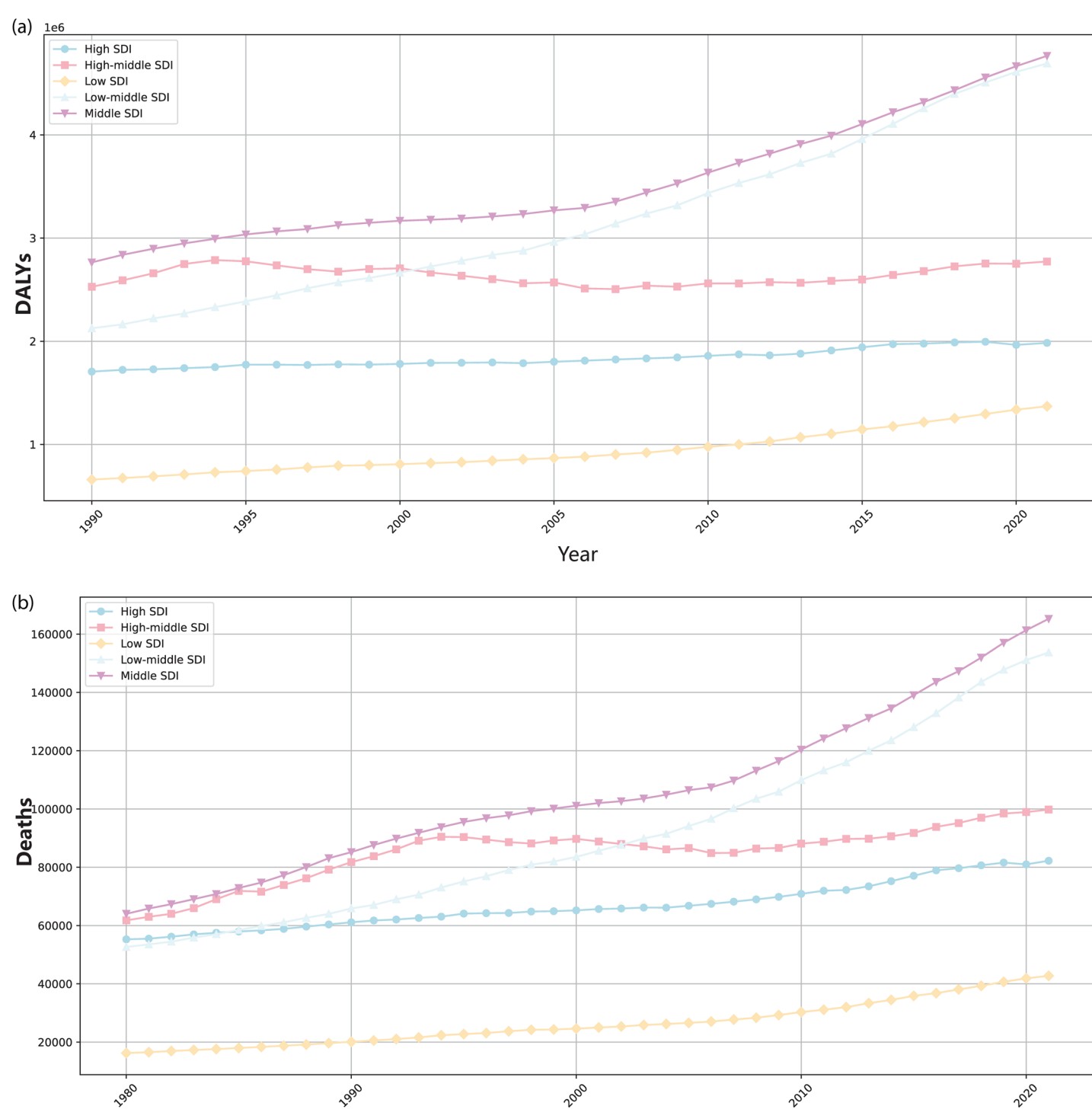

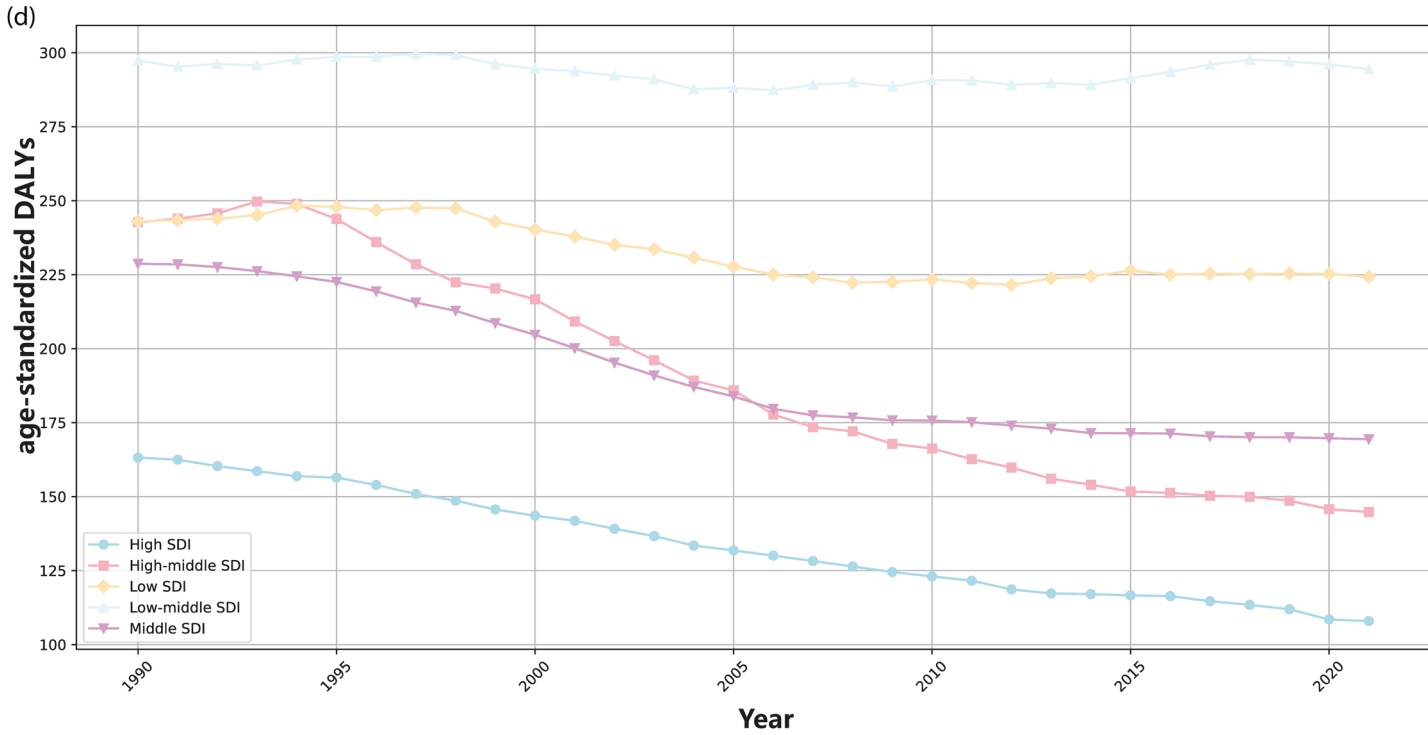

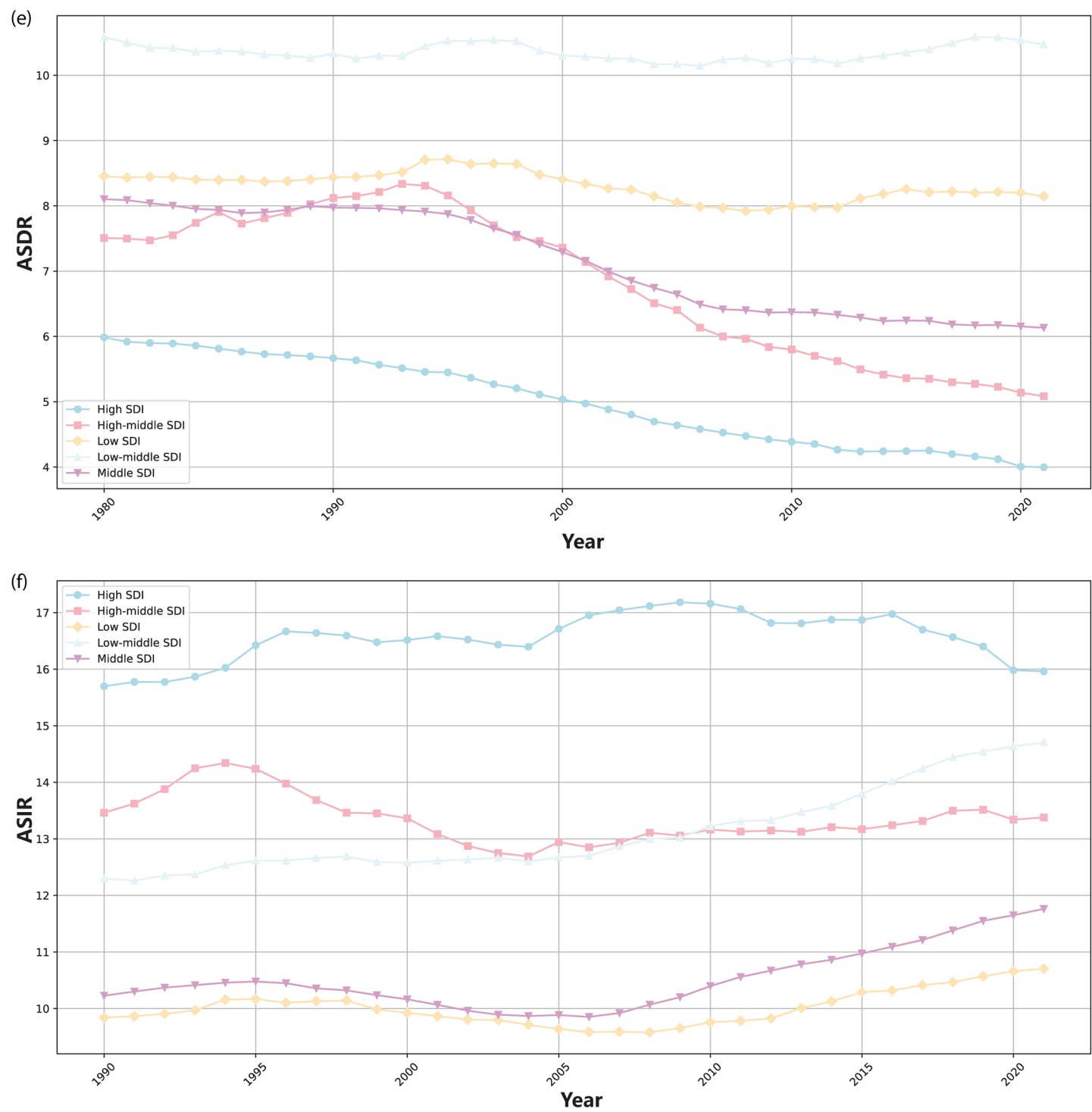

**Fig 5. Trends in the global disease burden of head and neck cancer (HNC) from 1980 to 2021, by sociodemographic index (SDI) regions.** (a) The trend in disability-adjusted life years (DALYs) of HNC across different SDI groups (high SDI, high-middle SDI, middle SDI, low-middle SDI, and low SDI). (b) The trend in deaths of HNC across different SDI groups. (c) The trend in incidence of HNC across different SDI groups. (d) The trend in age-standardized DALYs of HNC across different SDI groups. (e) The trend in age-standardized death rate (ASDR) of HNC across different SDI groups. (f) The trend in age-standardized incidence rate (ASIR) of HNC across different SDI groups.

**The global trend of burden of HNC by gender.** Table 1 displays that the EAPC of ASIR in females is 0.50 (95% CI: 0.45 – 0.55) more than the content in males of –0.09 ((95% CI: [–0.17]–[–0.01]) (Fig 4f). Simultaneously, the EAPC of ASDR is –0.56 ((95% CI: [–0.63]–[–0.50]) in females and –0.73 ((95% CI: [–0.81]–[–0.66]) in males (Fig 4e). However, male patients show higher values in incidence, mortality, DALYs, ASIR, ASDR, and age-standardized DALYs relative to female patients (Fig 4a–4f)).

**The global trend of burden of HNC by age.** The trends of incidence, deaths, DALYs for HNC across different age groups (5-year intervals) from 1980 to 2021 are presented in Table 2. In 2021, compared with 1990, there is an overall increasing trend in the incidence, deaths, and DALYs for HNC with increasing age, except for a decrease in deaths and DALYs among individuals aged under 20 (Fig 3a–3c). Since 2005, the DALYs in the 55-59 age group have surpassed those of the 50-54 age group, which can maintain the highest values in the subsequent years (Fig 3a). Meanwhile, its incidence has consistently ranked among the top two, along with the 60-64 age group (Fig 3c). In terms of mortality, the 65-69 age group has recently overtaken the 60-64 age group, becoming the group with the highest number of deaths (Fig 3b). When analyzing incidence and mortality trends, we find that the 70-74 age group has exhibited the fastest growth in both categories in recent years, requiring further attention (Fig 3b–3c).

## Prediction of the global trend of burden of HNC

After analyzing the global burden of HNC, research was performed to predict the future burden of HNC using our designed HNCP-T framework. In combination with the previous analysis, these predictions offer novel insights for the prevention and management of HNC.

**Prediction of the global trend of burden of HNC, from 2022 to 2030.** The ASIR is projected to increase annually from 2022 to 2030 (EAPC = 0.22, 95% CI: 0.09–0.35). By contrast, the ASDR and age-standardized DALYs are expected to decline during the same period, with EAPCs of –0.40 (95% CI: [–0.63]–[–0.16]) and –0.18 (95% CI: [–1.03]–[–0.59]), respectively (Table 3 and Fig 6a–6c).

**Prediction of the global trend of burden of HNC by Gender, from 2022 to 2030.** From 1990 to 2021, the ASIR of HNC shows an upward trend for both men and women, which is a trend that is projected to continue from 2022 to 2030 (Table 3 and Fig 7a and 7d). The EAPC for ASIR in women is 0.48 (95% CI: 0.26–0.71), which is more than double the growth rate observed in men, whose EAPC for ASIR is 0.20 (95% CI: 0.14–0.26 )(Table 3). However, the ASDR and age-standardized DALYs are expected to decline for both men and women from 2022 to 2030, with EAPCs of –0.54 (95% CI: [–0.61]–[–0.46]) and –0.92 (95% CI: [–1.14] –[–0.69]) , respectively (Table 3 and Fig 7b and 7e). In addition, the age-standardized DALYs rate is predicted to decrease slightly, with an EAPC of –0.14 (95% CI: [–0.83]–[–0.49]) for women and –0.66 (95%: [–0.83]–[–0.49]) for men (Table 3 and Fig 7c and 7f).

**Prediction of the global trend of burden of HNC by SDI, from 2022 to 2030.** From 2022 to 2030, global HNC burden trends show a strong correlation with socio-economic development. The disease indicators in high SDI areas are trending down (Fig 8a–8c), while disease indicators in low SDI areas are trending up (Table 3 and Fig 9e–9g). As illustrated in Figs 8 and 9, the forecasted trends in disease burden from 2022 to 2030 reveal significant discrepancy across regions with varying SDI levels. In high SDI regions, all three indicators are expected to decline, with estimated annual percentage changes (EAPCs) of –0.15, (95% CI: [–0.38]- 0.09), –1.08, (95% CI: [–1.16] – [–0.99]), and –1.08, (95% CI: [–1.16] – [–0.99]), respectively (Table 3 and Fig 8). However, in other regions, specifically high-middle SDI, middle SDI, low-middle SDI, and low SDI, an increasing trend in ASIR is expected from 2022 to

**Table 3. Predictions in global head and neck cancer (HNC) burden from 2022 to 2030 by gender (male and female) and 5 sociodemographic index (SDI) regions (high SDI, high-middle SDI, middle SDI, low-middle SDI and low SDI), along with the overall trend**

| Characteristics | ASIR | ASDR | Age-standardized DALY rate |
|---|---|---|---|
| Global | 0.22 (0.09, 0.35) | -0.4 (-0.63, -0.16) | −0.81 (−1.03, −0.59) |
| Sex | - | | |
| Female | 0.48 (0.26, 0.71) | −0.92 (−1.14, −0.69) | −0.14 (−0.3, 0.01) |
| Male | 0.2 (0.14, 0.26) | −0.54 (−0.61, −0.46) | −0.66 (−0.83, −0.49) |
| SDI | - | | |
| High SDI | −0.15 (−0.38, 0.09) | −1.08 (−1.16, −0.99) | −0.86 (−0.98, −0.74) |
| High-middle SDI | 0.22 (0.09, 0.35) | −1.82 (−1.97, −1.68) | −0.86 (−0.98, −0.74) |
| Middle SDI | 0.45 (0.14, 0.76) | −0.13 (−0.21, −0.05) | −0.16 (−0.21, −0.11) |
| Low-middle SDI | 1.39 (1.15, 1.63) | −0.19 (−0.39, 0.01) | −0.1 (−0.19, −0.01) |
| Low SDI | 0.39 (0.25, 0.54) | −0.06 (−0.1, −0.02) | −0.14 (−0.28, −0.01) |

2030 (Table 3 and Figs 8d–8f). Notably, the EAPC for ASIR is the highest in low-middle SDI regions (1.39, 95% CI: 1.15 – 1.63), followed by middle SDI regions (0.45, 95% CI: 0.14, 0.76) (Table 3 and Fig 8a, 8d, 8g and 9b, 9e)). Meanwhile, the ASDR and age-standardized DALY rate are expected to decline in these regions (Fig 9b, 9c, 9e, 9f. High-middle SDI regions notably exhibit the lowest EAPC for ASDR at –1.82 (95% CI: $[-1.97], [-1.69]$) and for age-standardized DALYs at –0.86 (95% CI: $[-0.98], [-0.74]$), followed by high SDI regions with EAPCs of –1.08 (95% CI: $[-1.16], [-0.99]$) for ASDR and –0.86 (95% $CI$: $[-0.98], [-0.74]$) for age-standardized DALYs, respectively (Fig 9a, 9d, 9e).

**Comparative experiment with other models.** Given that LSTM [23] and linear regression [24] models are also often adopted to predict the global disease burden of cancer, to further validate the superiority of our proposed HNCP-T, we conduct a comparative experiment with Linear Regression and LSTM. The experimental setup is as follows:

- **Model Selection:** Linear Regression is chosen as a representative classical statistical model, and LSTM is selected as an alternative deep learning-based sequence model.
- **Training Process:** Each model is trained using the training dataset for 100 epochs.
- **Evaluation Metrics:** MAE is applied as the primary evaluation metric to assess the performance of each model based on the test dataset across three dimensions: ASIR, ASDR, and Age-Standardized DALY rate.
- **Comparison Analysis:** The MAE values for each model are compared to evaluate their predictive performance.

As shown in Table 4, the proposed HNCP-T model consistently achieves the lowest MAE values across all three metrics compared to Linear Regression and LSTM models. Specifically, the HNCP-T model records an MAE of 0.028 for ASIR, 0.018 for ASDR, and 0.015 for Age-Standardized DALY rate, outperforming the other models by a significant margin. These results demonstrate the HNCP-T model's superior ability to capture complex nonlinear relationships and temporal dependencies inherent in the HNC burden data, particularly in handling time-series signals and categorical data effectively.

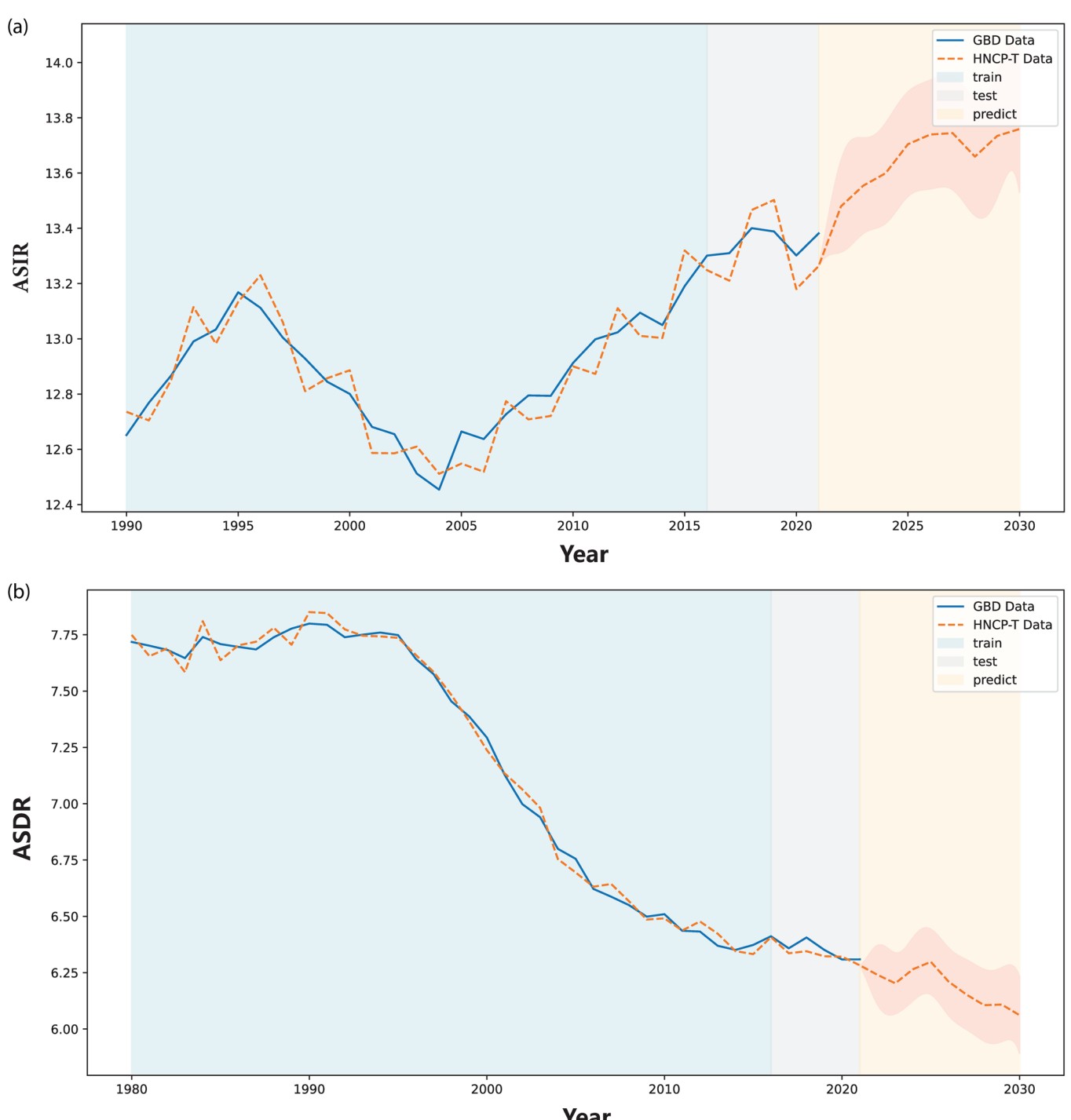

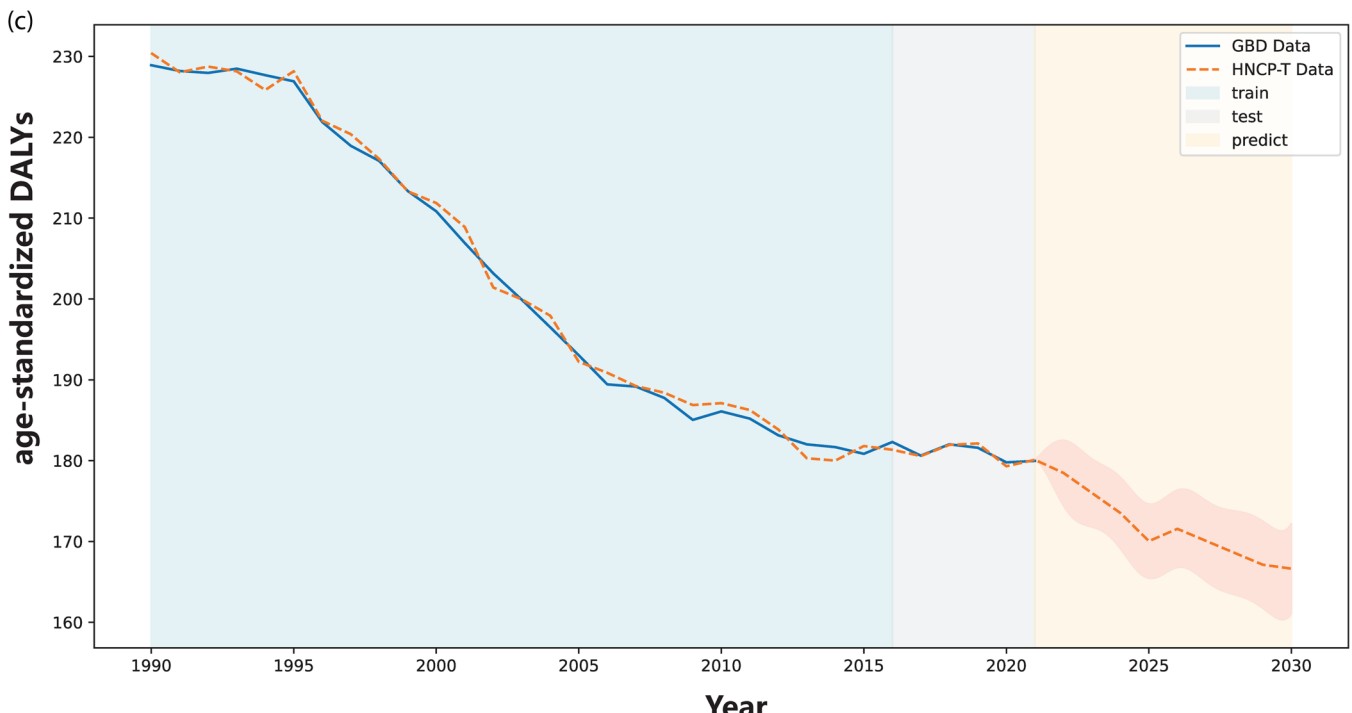

**Fig 6.** The rediction result of the global disease burden (GBD) of head and neck cancer (HNC) from 1980 to 2021. (a) - (c) display the comparison between the age-standardized incidence rate (ASIR), age-standardized death rate (ASDR), and age-standardized disability-adjusted life years (DALYs) values obtained using the proposed HNCP-T strategy and the actual data from the GBD dataset. As presented in these figures, the blue shaded area represents the comparison between the HNCP-T training results and the true GBD dataset labels from 1980 to 2015, the gray shaded area indicates the comparison between the HNCP-T test results and the true GBD dataset labels from 2016 to 2021, and the yellow shaded area denotes the comparison between the HNCP-T test results and the true GBD dataset labels projected for 2022 to 2030.

The enhanced performance of the HNCP-T model can be attributed to its self-attention mechanisms, which allow for better modeling of long-range dependencies and interactions within the data. Additionally, the robustness of the HNCP-T model in managing missing and noisy data further solidifies its advantage over traditional and other deep learning-based approaches in this context.

## Human participants

No human participants are involved in this study.

### Patient and public involvement

There is no involvement of patients or the public in the study's design. Moreover, no patients are recruited or included in the study.

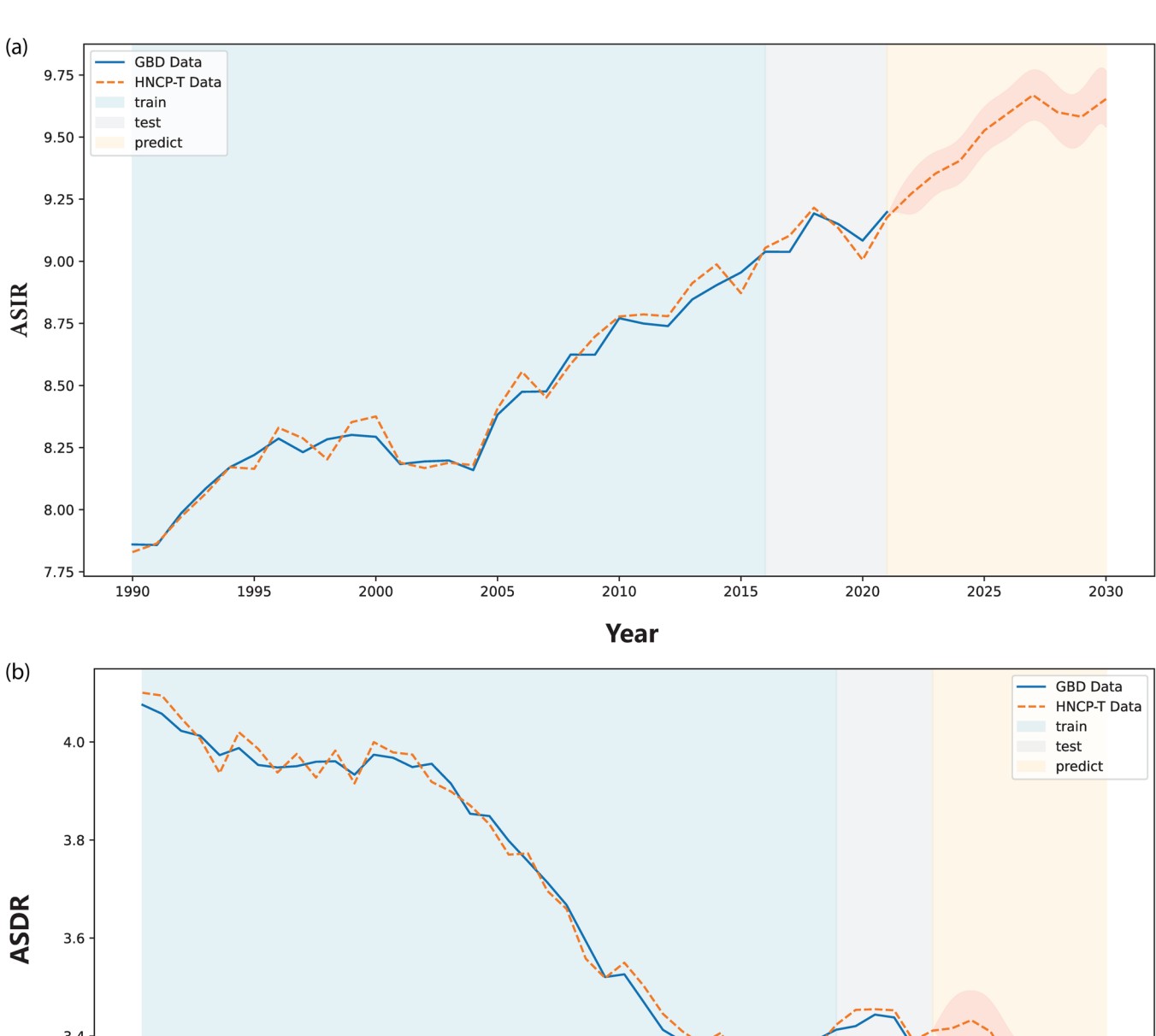

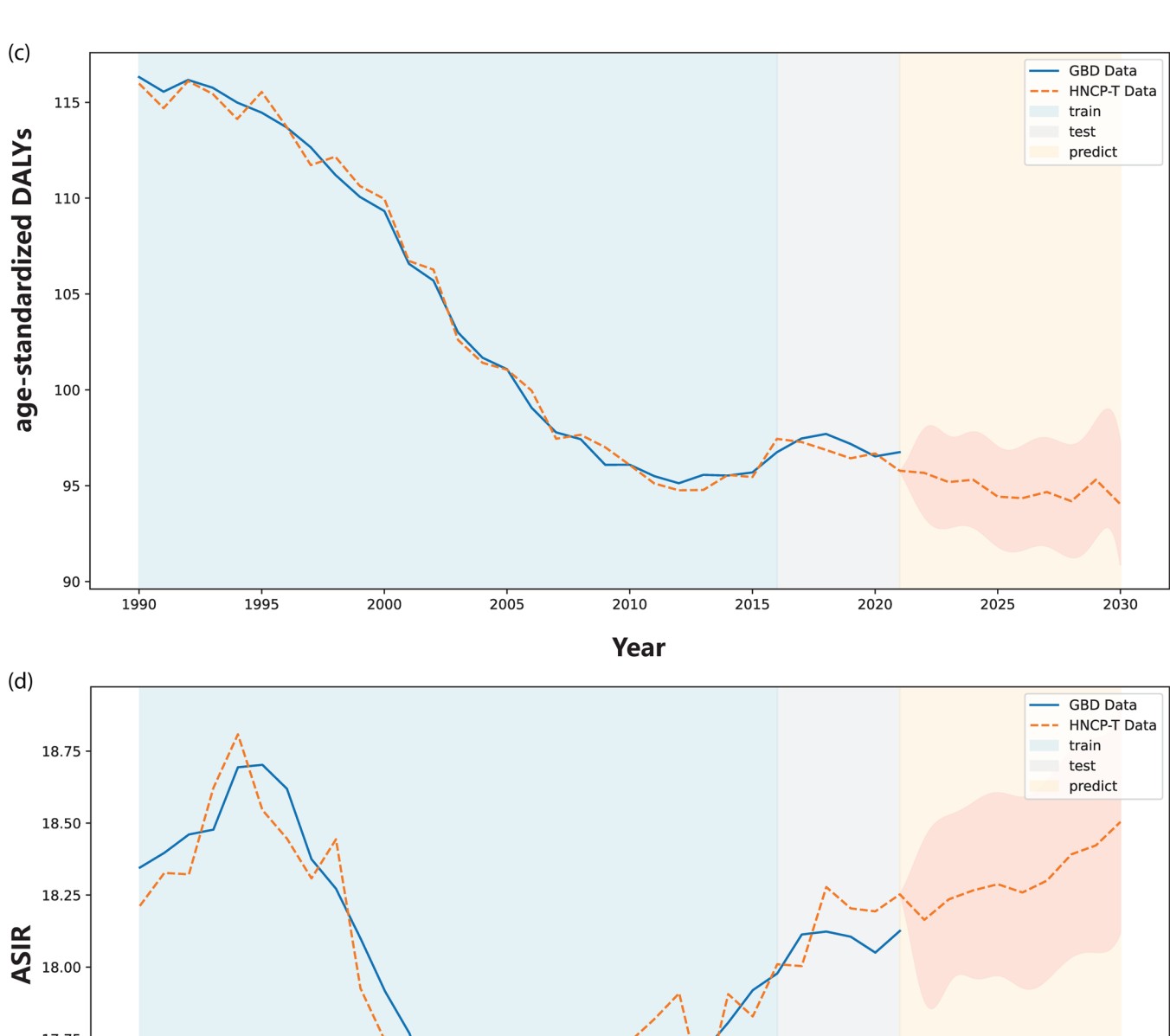

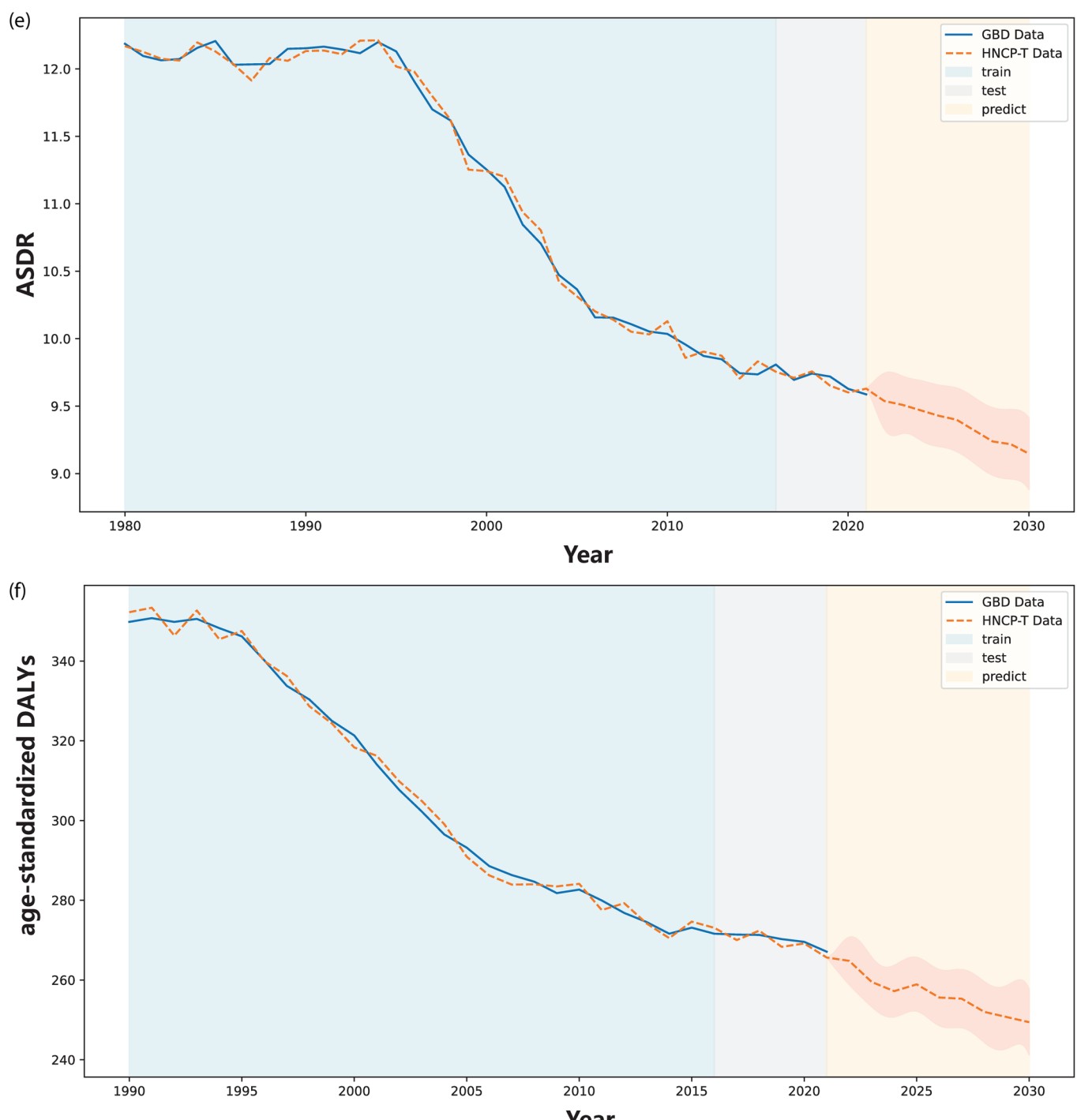

**Fig 7. The prediction result of the global disease burden (GBD) of head and neck cancer (HNC) from 1980 to 2021, by gender.** (a) to (c) show the comparison between the predicted and actual GDB data for females, covering ASIR, ASDR, and age-standardized disability-adjusted life years (DALYs), respectively, using the HNCP-T strategy. (d) to (f) present the same comparisons for males. As shown in these figures, the blue area represents the comparison between the training results of HNCP-T and actual GDB data from 1980 to 2015, the gray area covers the comparison between test results and actual data from 2016 to 2021, and the yellow area represents predictions for 2022 to 2030.

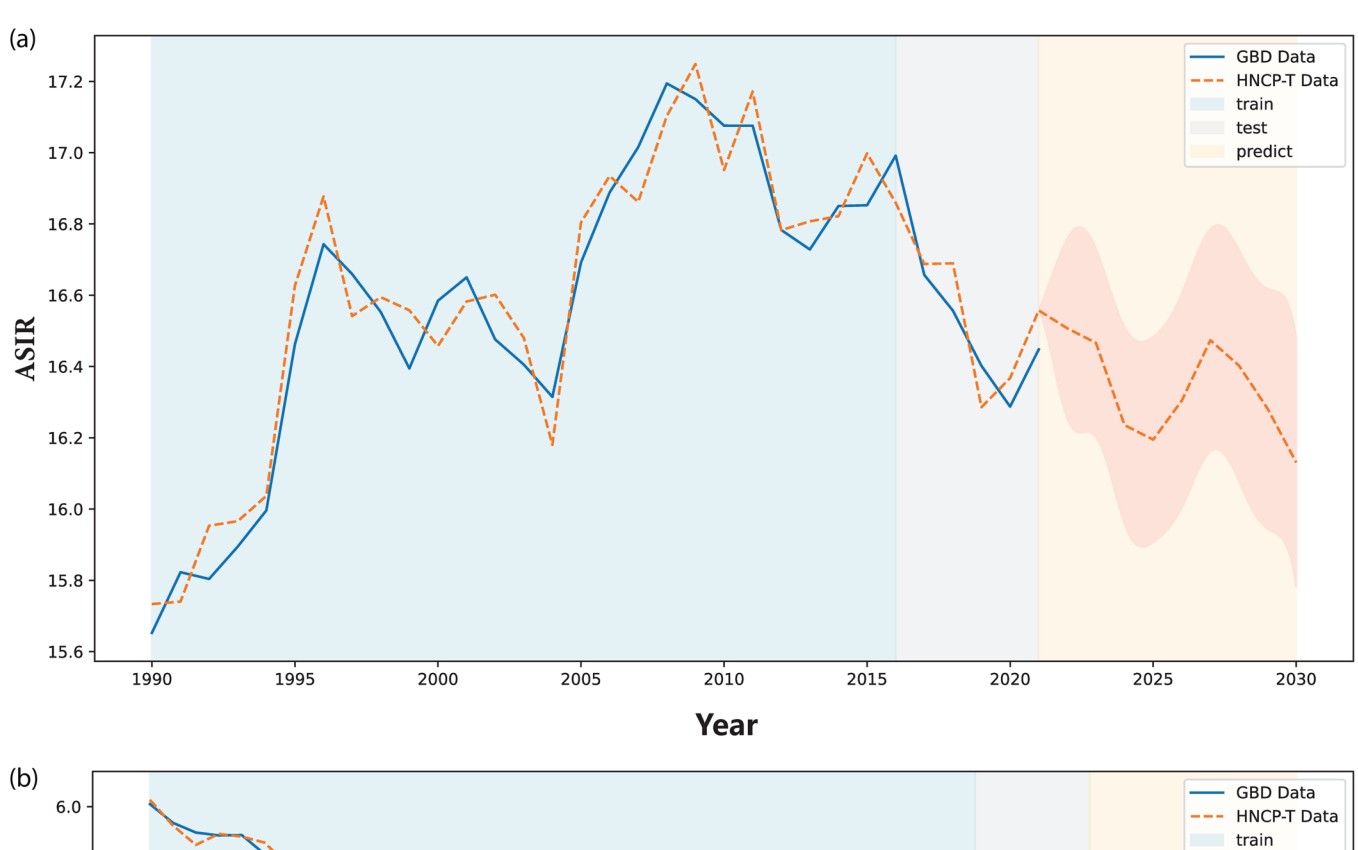

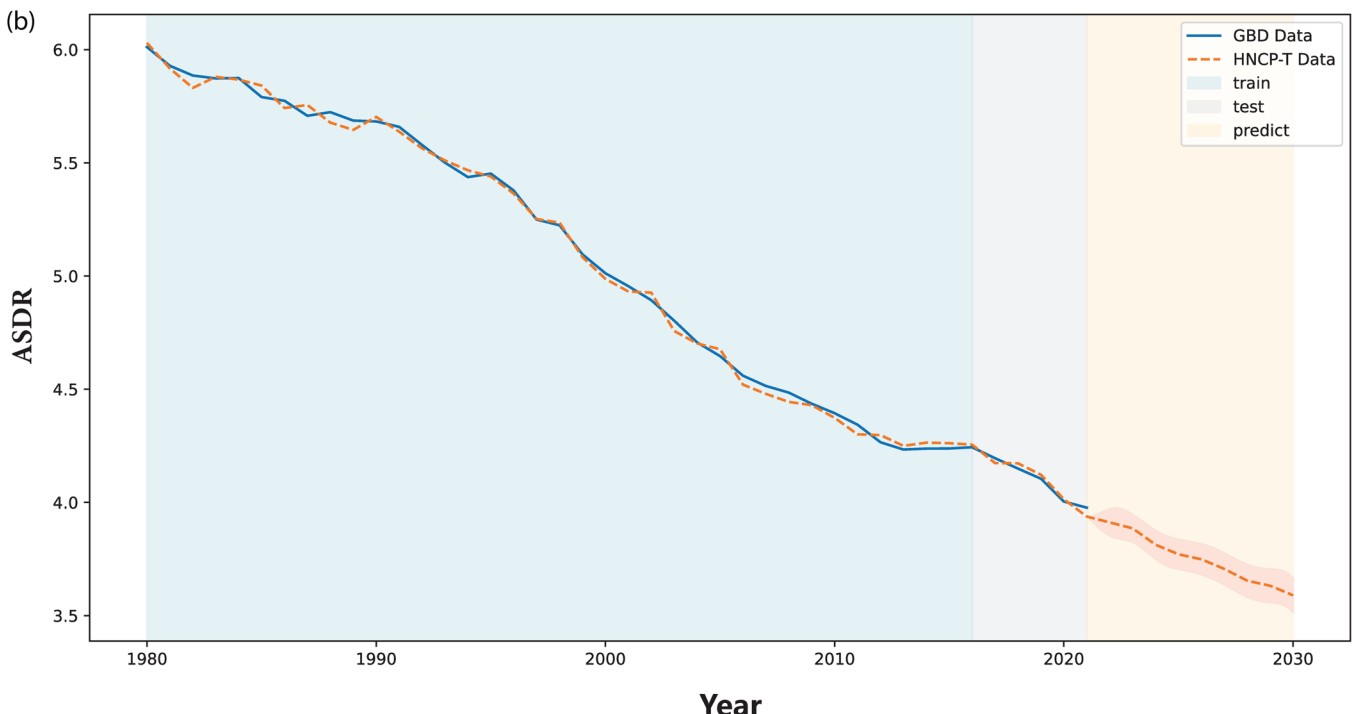

(c)

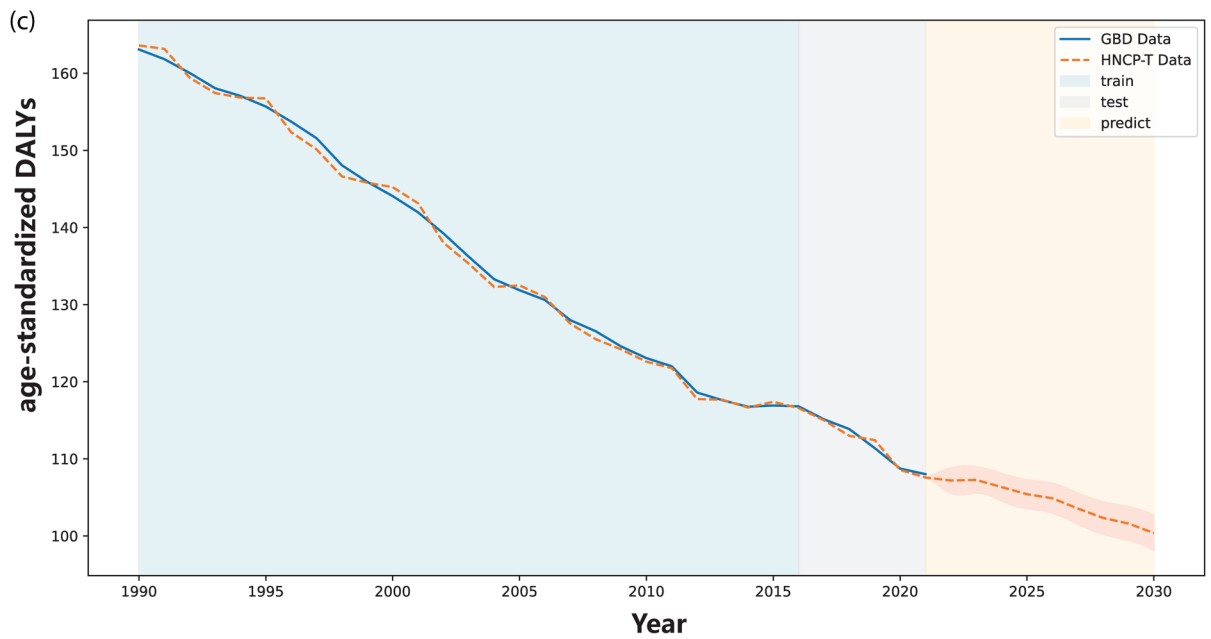

(d)

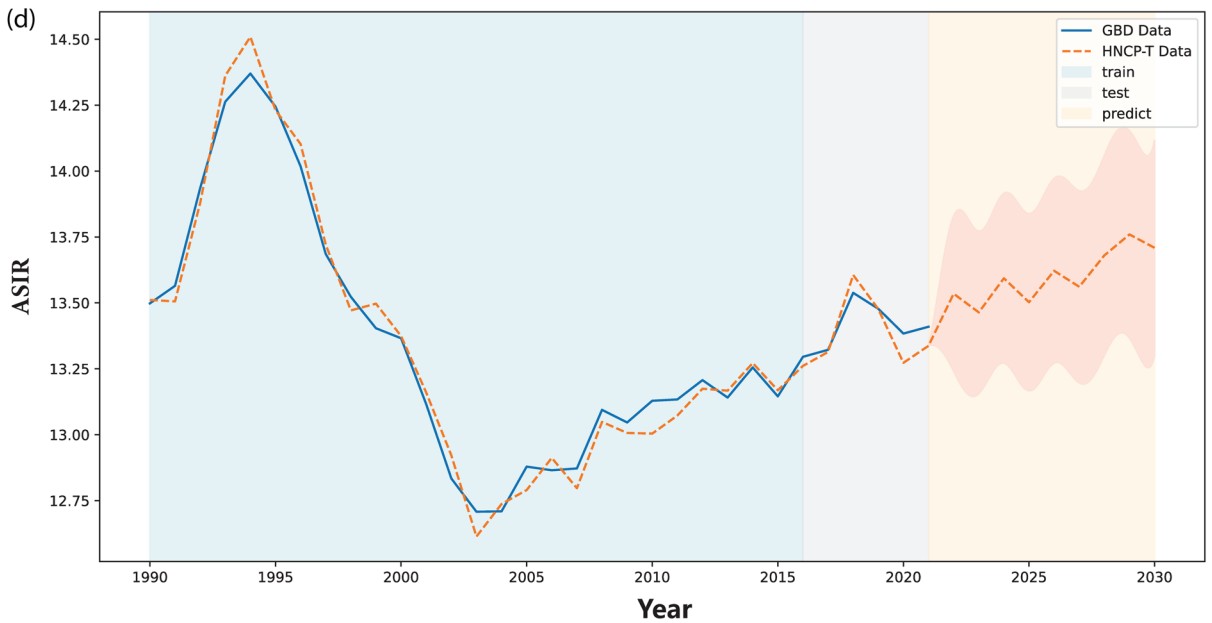

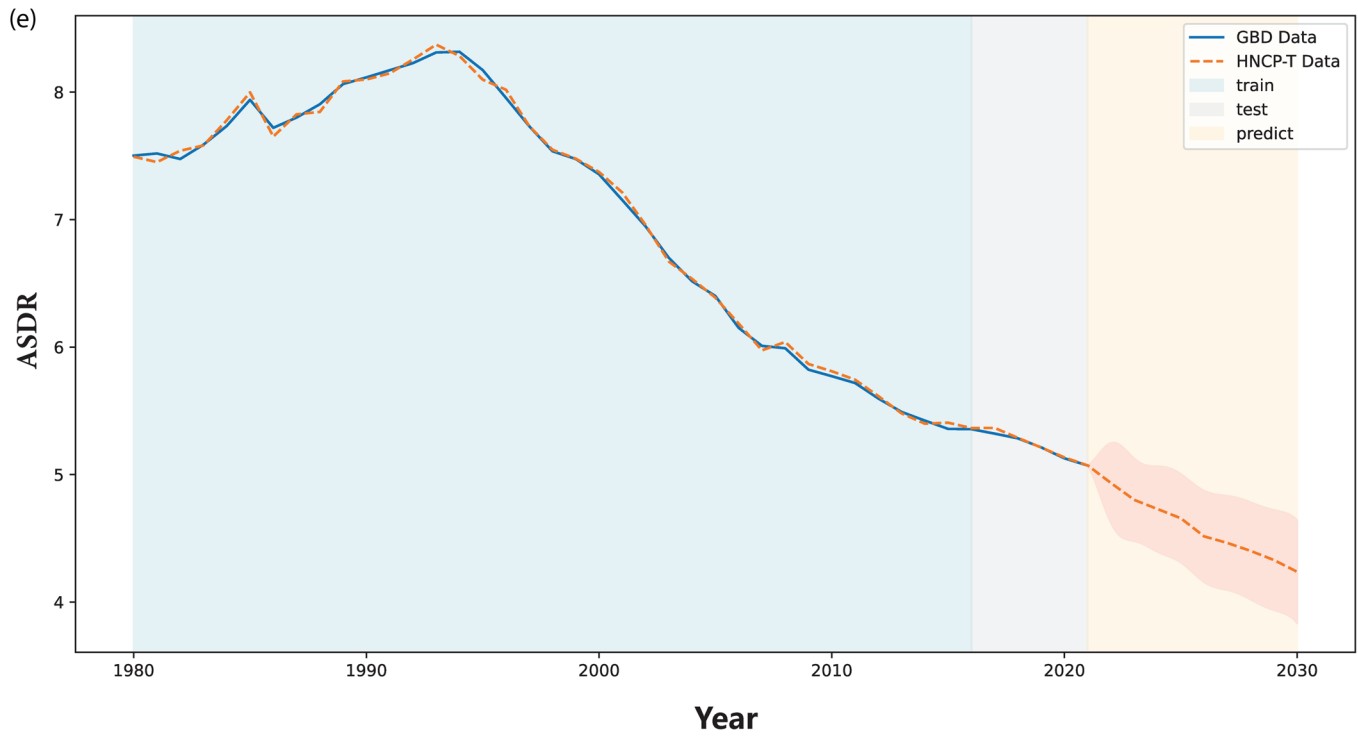

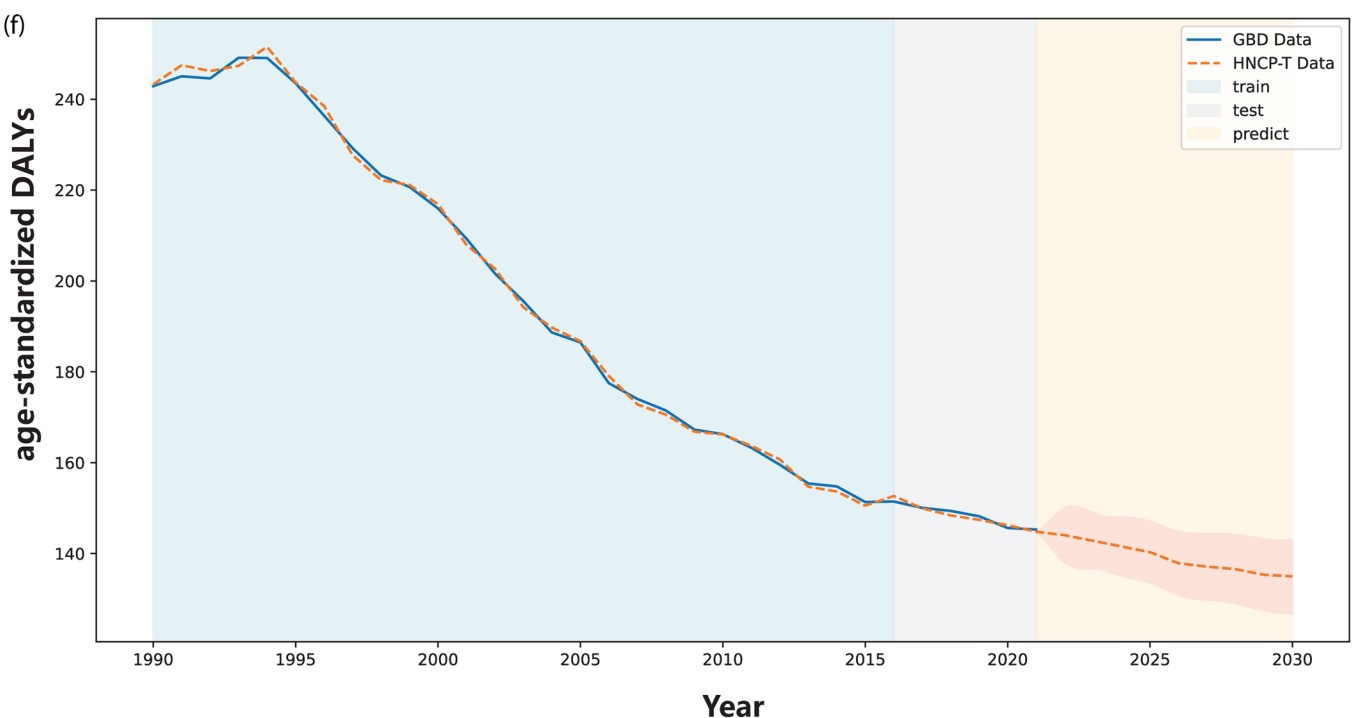

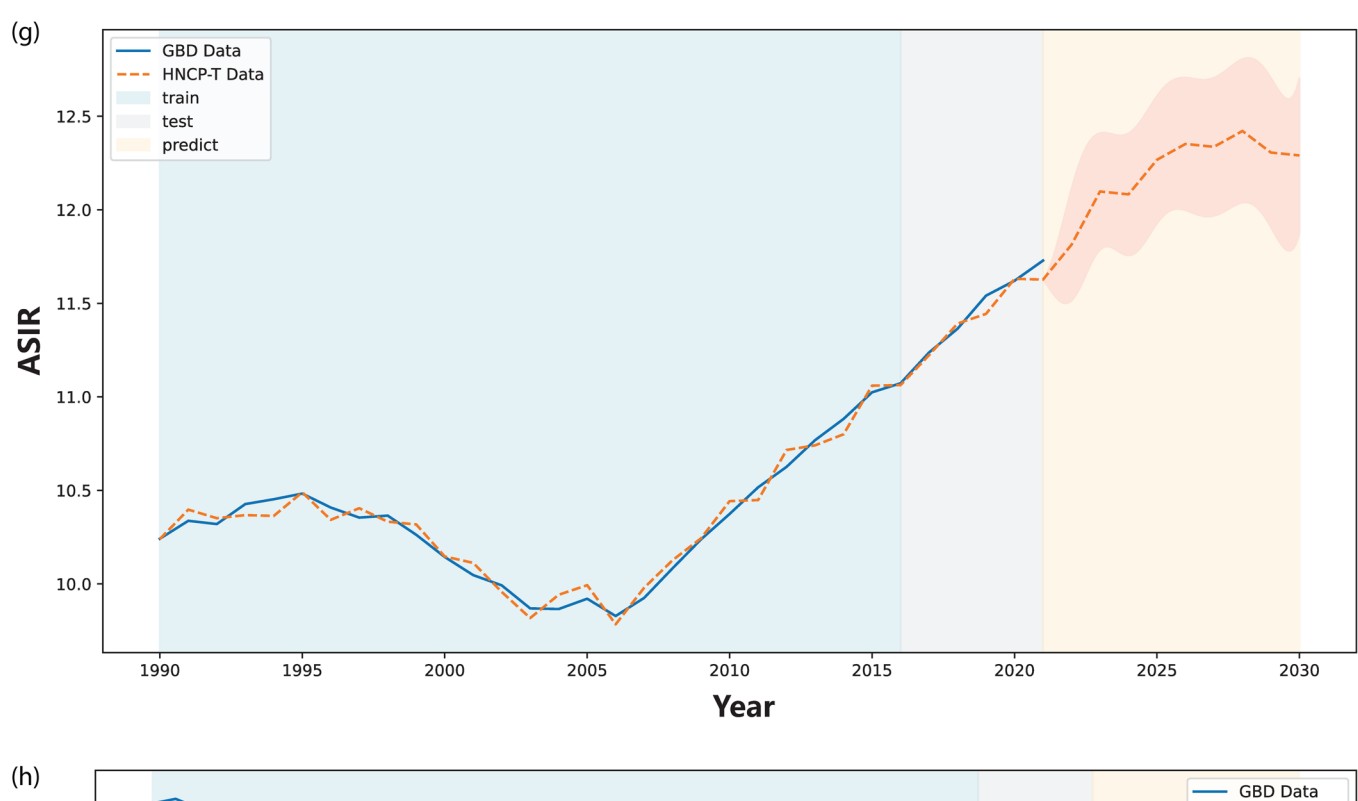

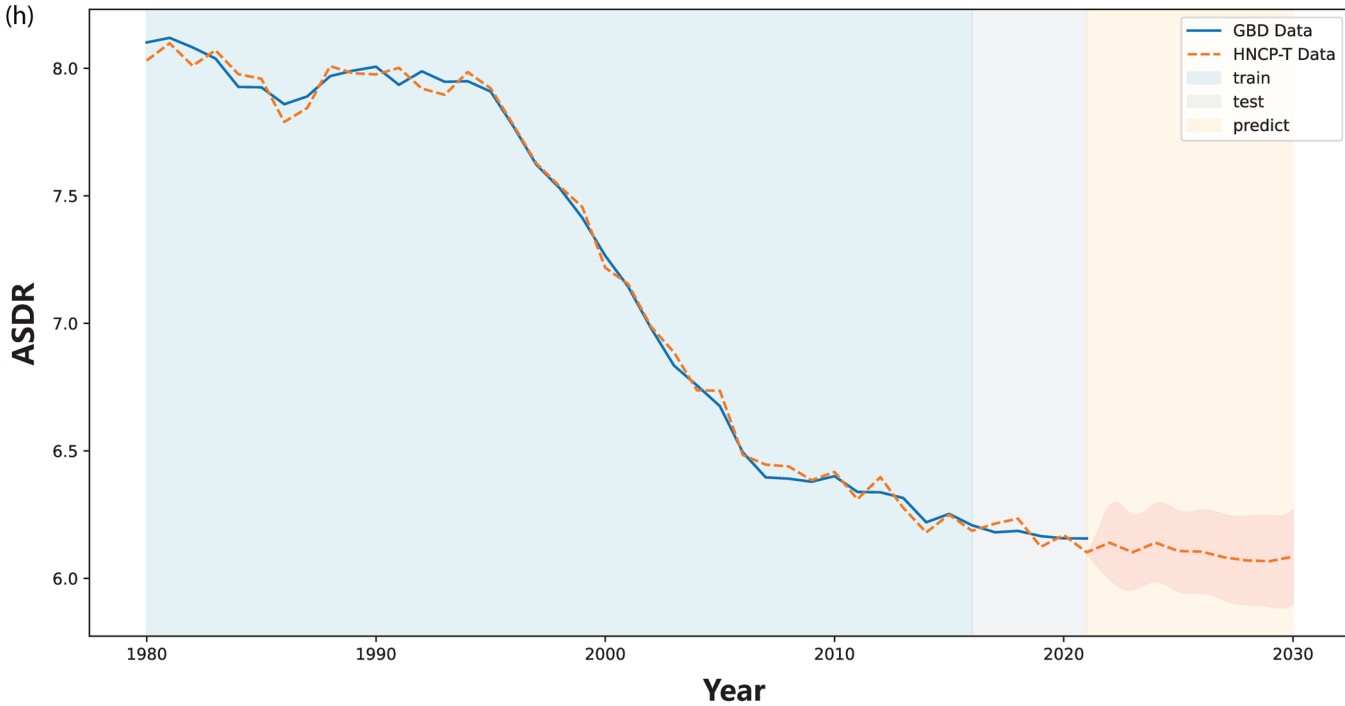

**Fig 8. The prediction result of the global disease burden (GBD) of head and neck cancer (HNC) from 1980 to 2021, by sociodemographic index (SDI) (part 1: High SDI, High middle SDI and Middle SDI).** (a) to (c) illustrate the comparison between the predicted and actual GBD data for high SDI, covering age-standardized incidence rate (ASIR), age-standardized death rate (ASDR), and age-standardized disability-adjusted life years (DALYs), respectively, using the HNCP-T strategy. (d) to (f) present the same comparisons for high-middle SDI. (g) to (h) show the comparison between the predicted and actual GBD data for middle SDI, covering ASIR and ASDR, using the HNCP-T strategy. As presented in these figures, the blue area indicates the comparison between the HNCP-T training results and actual GBD data from 1980 to 2015, the gray area covers the comparison between test results and actual data from 2016 to 2021, and the yellow area represents predictions for 2022 to 2030.

(a)

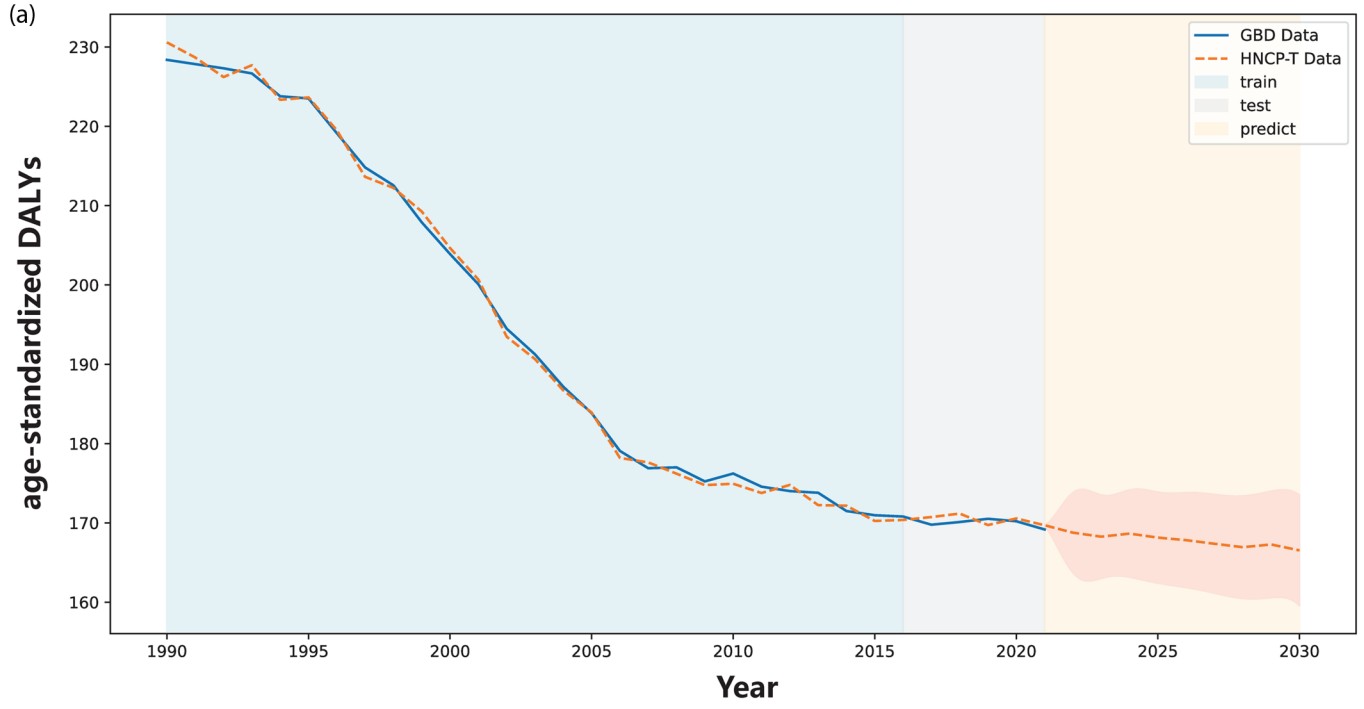

(b)

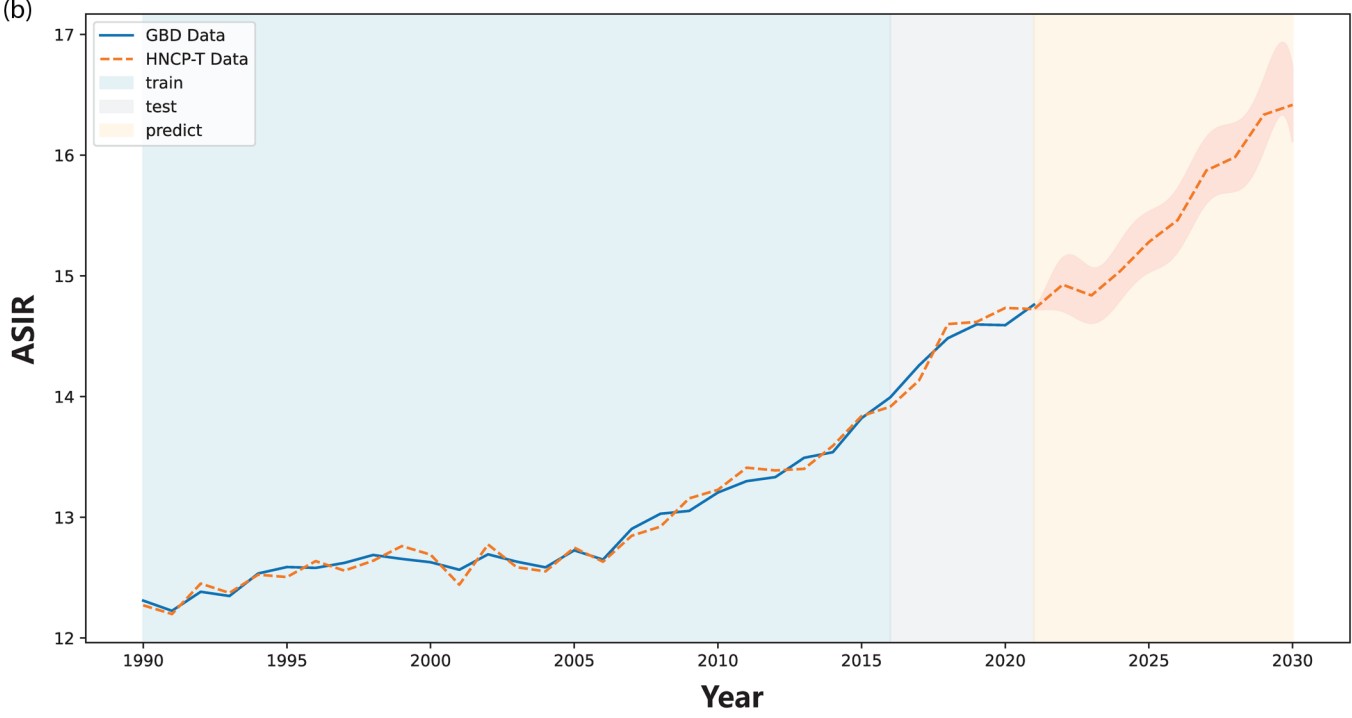

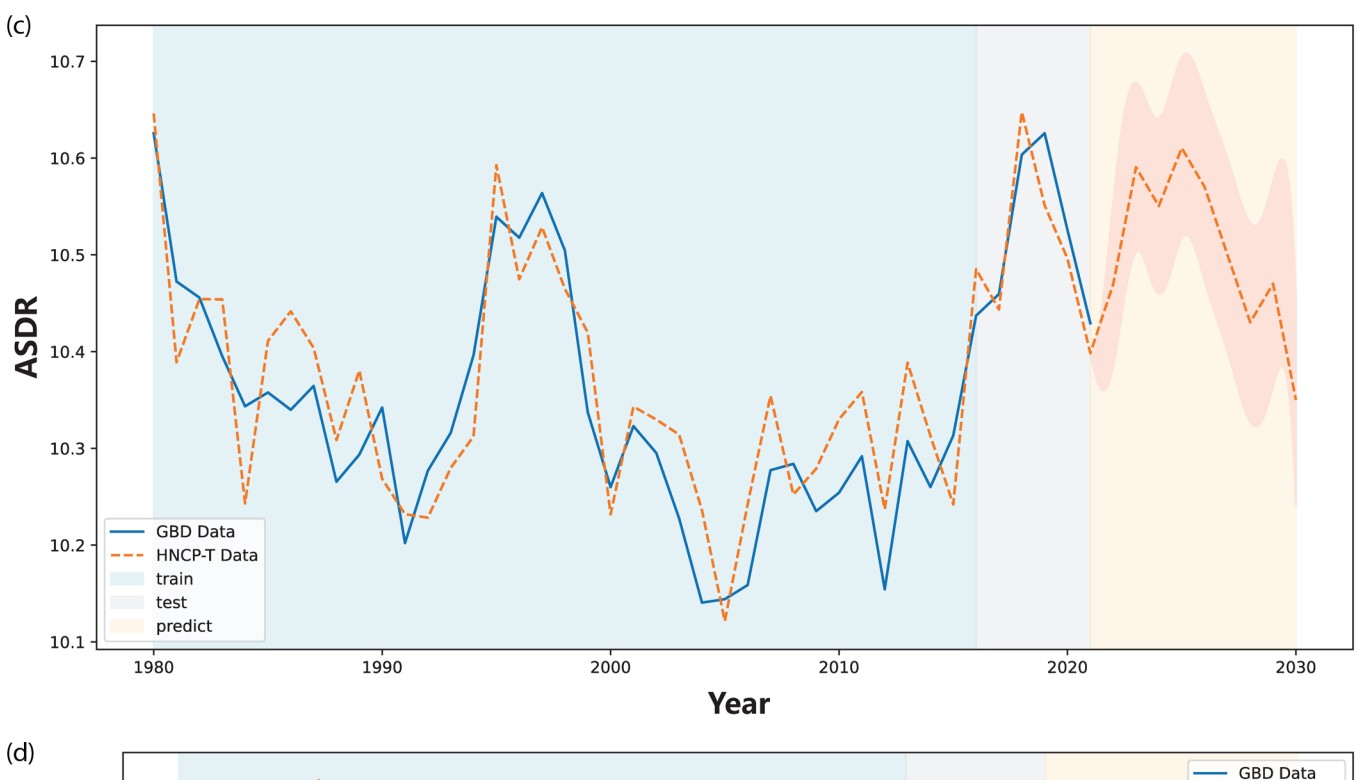

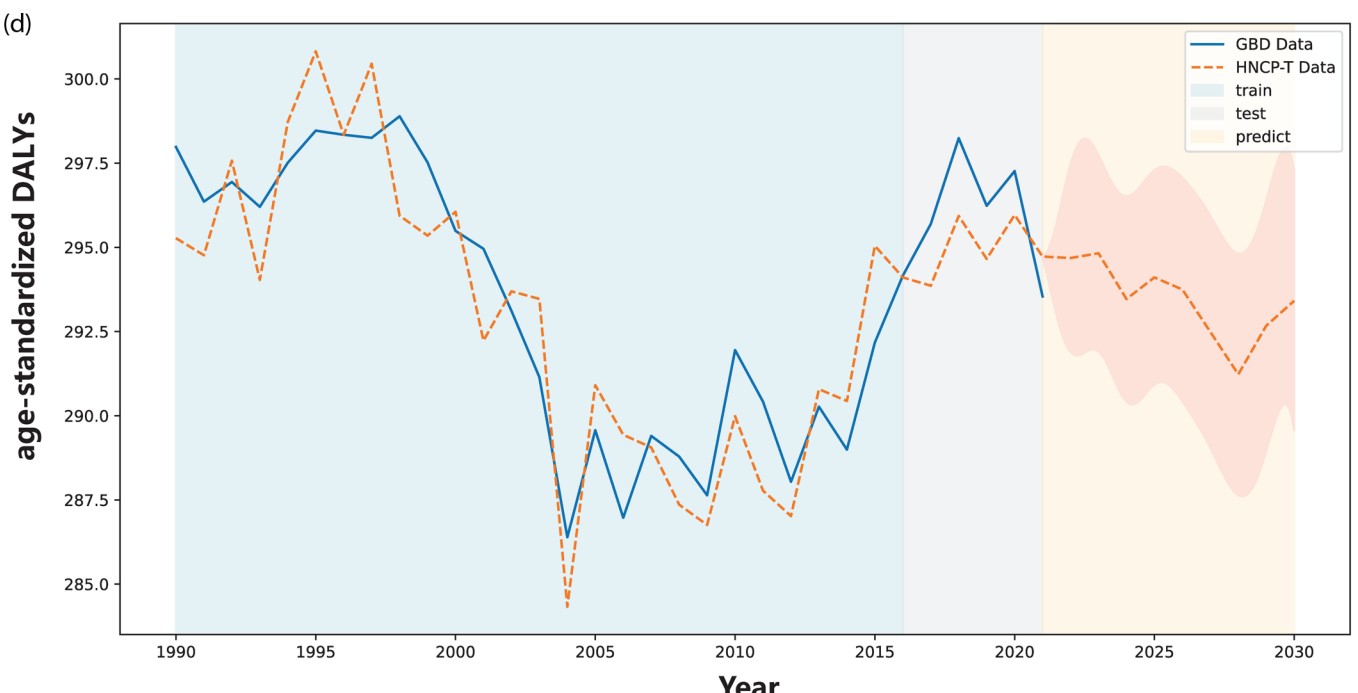

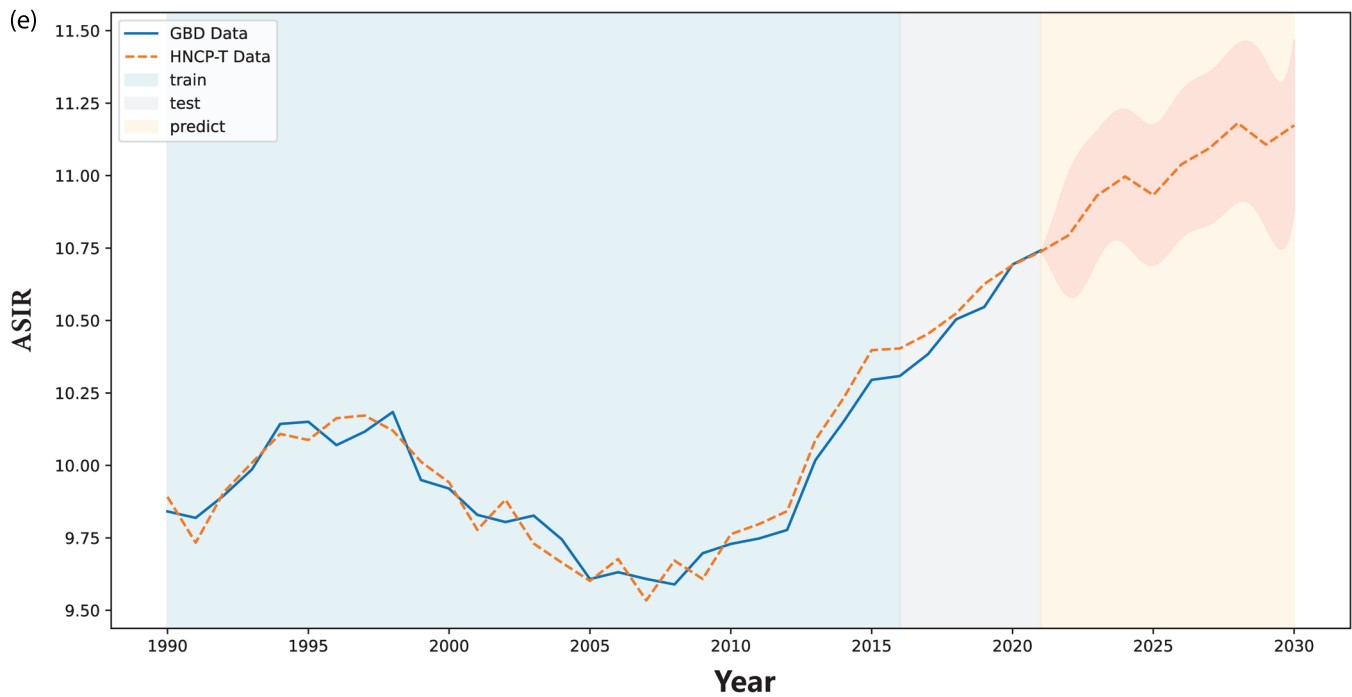

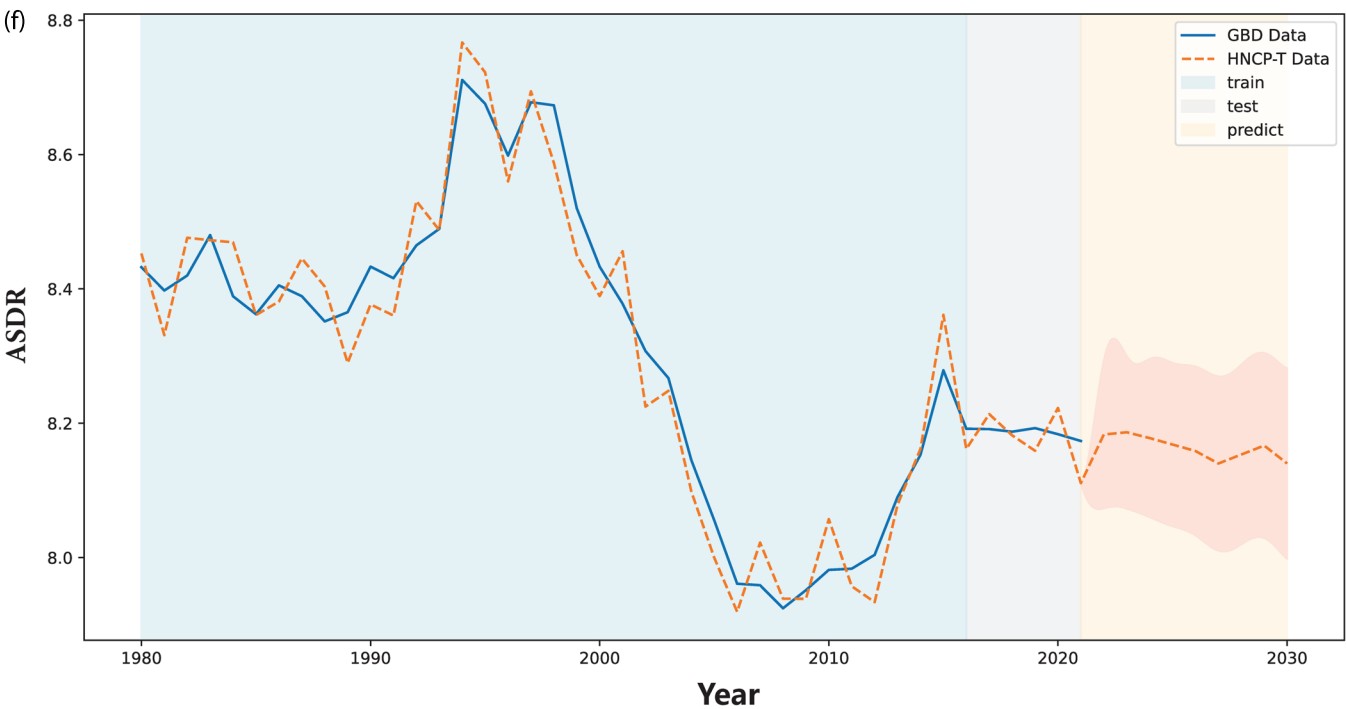

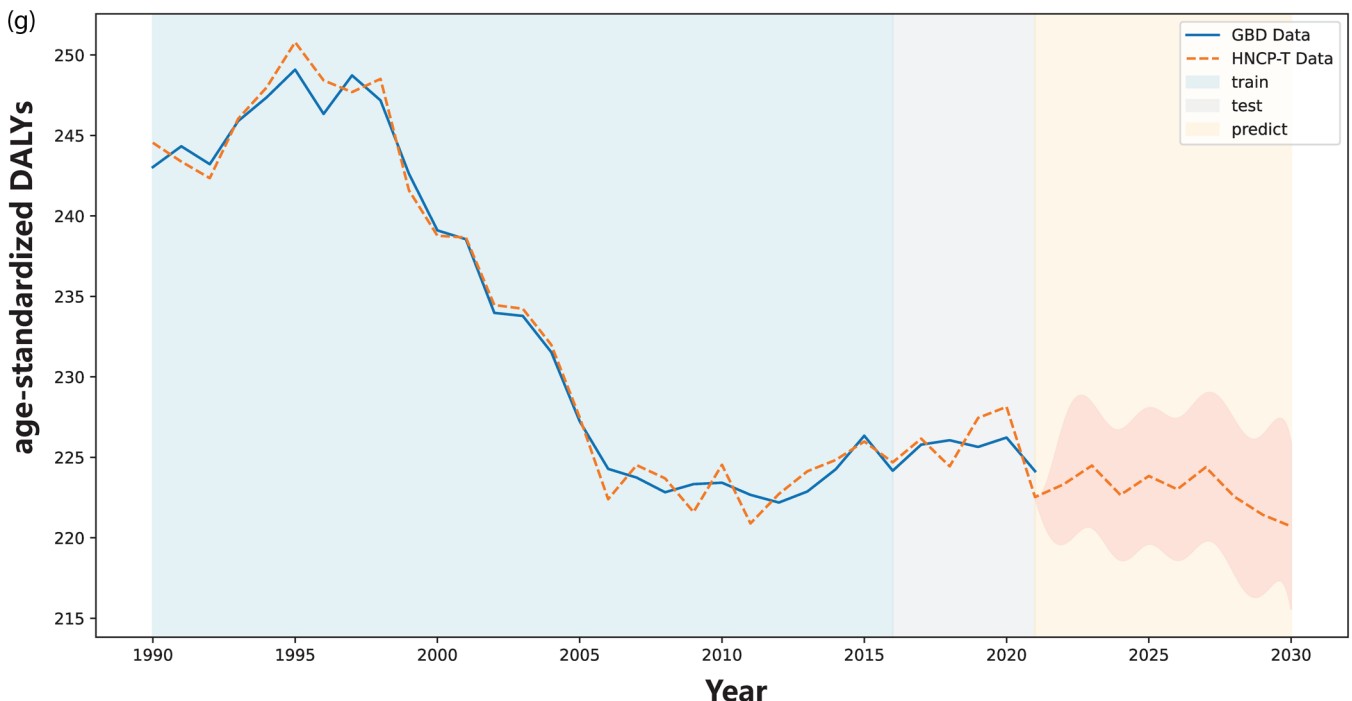

**Fig 9. The prediction result of the global disease burden (GBD) of head and neck cancer (HNC) from 1980 to 2021, by sociodemographic index (SDI) (part 2: Middle SDI and Low middle SDI).** (a) illustrates the comparison results in age-standardized DALYs between the predicted and actual GDB data for middle SDI, using the HNCP-T strategy. (b) to (d) present the comparison between the predicted and actual GDB data for low middle SDI, covering age-standardized incidence rate (ASIR), age-standardized death rate (ASDR) and age-standardized disability-adjusted life years (DALYs), respectively, using the HNCP-T strategy. (d) to (f) present the same com-parisons for high middle SDI. (e) to (g) indicate the comparison between the predicted and actual GDB data for low SDI, covering ASIR, ASDR and age-standardized DALYs respectively, using the HNCP-T strategy. In these figures, the blue area denotes the comparison between the training results of HNCP-T and actual GDB data from 1980 to 2015, the gray area covers the comparison between test results and actual data from 2016 to 2021, and the yellow area denotes predictions from 2022 to 2030.

**Table 4. Comparison of mean absolute error (MAE) for ASIR, ASDR, and age-standardized DALY rate across different models**

| Model | ASIR MAE | ASDR MAE | Age-standardised DALY rate MAE |
|---|---|---|---|
| Linear Regression | 0.045 | 0.030 | 0.025 |
| LSTM | 0.035 | 0.022 | 0.020 |
| **HNCP-T** | **0.028** | **0.018** | **0.015** |

## Discussion

In this study, we provide novel insights into the global burden of HNC based on deep learn-ing, specifically a transformer-based model. The transformer-based model offers significant advantages over traditional sequence models such as LSTM and RNNs, including the abil-ity to capture long-range dependencies more effectively through self-attention mechanisms and enable parallel processing, which lead to improved performance in forecasting tasks. Our comparative experiments further validate that the transformer model achieves lower MAE values across key metrics, underscoring its superior capability in handling complex nonlin-ear relationships and temporal dependencies inherent in the global burden of head and neck cancers. Additionally, the transformer model demonstrates greater robustness in managing missing and noisy data, which is crucial for accurate disease burden predictions in diverse

and heterogeneous datasets. We analyze historical trends in HNC incidence, mortality, and DALYs, along with future projections for the age-standardized rates of these metrics. Our results underscore key patterns in disease burden across gender, socioeconomic regions, and age groups, as well as the effects of modern screening and treatment approaches.

## Increasing incidence driven by environmental and behavioral factors

Our results emphasize a consistent increase in the age-standardized incidence rate (ASIR) of HNC from 1980 to 2021, with predictions indicating a further 1.3-fold rise by 2030. This increase in incidence is consistent with growing exposure to risk factors such as smoking, alcohol consumption, and human papillomavirus (HPV) [10,11]. The increase in risky behaviors, such as smoking and alcohol consumption, are potential factors in the rising incidence of HNC [25]. Studies have shown that from 1990 to 2017, global adult per capita alcohol consumption increased from 5.9 L (95% CI: 5.8–6.1) to 6.5 L (95%CI: 6.6–6.9), and it is projected to reach 7.6 L (95% CI: 6.5–10.5) by 2030 [26]. The number of smokers have increased significantly from 1980 to 2021 [27,28]. As smoking and alcohol consumption are major risk factors for HNC, public health policies should focus on implementing targeted interventions to reduce these behaviors. This includes increasing awareness, providing support for cessation programs, and enforcing regulations to limit the availability of tobacco and alcohol. Furthermore, the rising global prevalence of HPV, which is also associated with cervical and anal cancers [29,30], is particularly alarming. The growing prevalence of HPV-re lated subsites in HNC is a crucial factor driving the significant changes observed in global HNC trends [31]. Expanding vaccination programs globally, particularly in regions with low coverage, could significantly lower this burden [32].

The role of advancements in diagnostic technologies is considered another significant factor. Narrow-band imaging (NBI) and other early detection techniques have significantly enhanced HNC screening outcomes [33]. Nevertheless, despite these technological improvements, regions with limited access to healthcare continue to exhibit rising incidence rates. Moreover, future studies should concentrate on exploring how further innovation in screening, combined with wider global access, can mitigate the growing incidence.

## Declining mortality and DALYs: the role of treatment advancements

While HNC incidence continues to increase, it is suggested in our projections that the age-standardized death rate (ASDR) and DALYs will stabilize or decrease by 2030. This divergence between rising incidence and declining mortality is primarily caused by advances in treatment strategies, including the combined use of surgery, radiotherapy, chemotherapy, and targeted therapies such as Epidermal Growth Factor Receptor (EGFR) inhibitors [34]. Patient outcomes have been further enhanced by improved genetic diagnostics and individualized treatment plans [35].

The predicted decrease in ASDR and DALYs shows the positive impact of modern treatments, while continued monitoring is vital to ensure that these trends persist across different regions and population groups. It is essential for future studies to concentrate on expanding access to these advanced treatment modalities in regions with limited healthcare infrastructure.

## Gender-based trends in HNC burden

A key finding is the distinct gender-based trends in HNC burden. Even though men continue to exhibit higher overall ASIR, ASDR, and DALYs compared with women, the incidence

rate among women is increasing more rapidly [36,37]. The annual incidence rate in females is expected to increase at a rate of 0.50 (95% CI: 0.45-–0.55), whereas the rate is reducing slightly at –0.09 in males (95% CI: [–0.17]–[–0.01]). Despite the rise in incidence, females show a smaller decrease in mortality rates compared with males [38]. These gender disparities are probably caused by several factors, such as behavioral differences, hormonal influences, and lifestyle choices.

While hormonal factors, such as estrogen's potential protective role against HNC, have been extensively studied, they do not fully explain the faster rise in incidence among women [36]. Further research is needed to investigate other hormonal mechanisms, such as the role of menopause and hormone replacement therapy in modifying HNC risk. Lifestyle factors, including alcohol consumption and tobacco use, are also evolving differently by gender, with females exhibiting a slower decline in alcohol use and lower rates of tobacco use globally [39]. Notably, alcohol consumption patterns among women, particularly in higher-income regions, are rising, which may contribute to the increasing HNC incidence. Moreover, evolving trends in smoking cessation between men and women might explain some of the disparities in disease progression.

Recurrence and second primary cancers (SPCs) are significant challenges to long-term survival, especially in women. The increase in late-stage HNC among females indicates that gender-based interventions may be necessary to deal with the above-mentioned risks. For example, educational campaigns targeting alcohol consumption among women, or screenings tailored to address late-stage diagnoses could be pivotal in mitigating the rising burden among women. Moving forward, to effectively mitigate the growing HNC burden among women, public health strategies must incorporate these gender differences in risk factors and disease progression.

## Socioeconomic disparities in HNC burden

Our findings show substantial disparities in HNC burden across regions with different SDI. High SDI regions reveal a reducing trend in ASIR, ASDR, and DALYs, reflecting advancements in healthcare access and preventive measures. In contrast, low- and middle-SDI regions are expected to bear a greater disease burden by 2030, with middle-SDI regions being projected to confront the highest overall HNC impact [40,41].

Socioeconomic factors, such as income level, education, and healthcare infrastructure, play a significant role in shaping these disparities. For instance, income inequality is associated with limited access to early screening and treatment, leading to delayed diagnoses and poorer outcomes. In regions with lower education levels, public awareness about HNC risk factors—such as tobacco use, alcohol consumption, and HPV vaccination—tends to be lower, further contributing to rising incidence and mortality rates. These regional disparities emphasize the need for targeted healthcare interventions. Improved access to screening and treatment in low- and middle-income regions exerts a critical role in reversing these trends. For example, socioeconomic indicators such as poverty rates, literacy levels, and healthcare expenditure per capita are closely correlated with HNC incidence and mortality rates. Regions with higher poverty and lower healthcare spending usually face more advanced disease stages at diagnosis, resulting in worse prognosis and survival rates. Developing cost-effective, scalable diagnostic tools and expanding vaccination programs could notably alleviate the burden in these areas. In addition, global efforts need to be focused on reducing the delay in diagnosis, remaining a major issue in low-SDI regions [41]. Addressing these socioeconomic barriers through public health policies aimed at improving health literacy, increasing vaccination coverage, and providing affordable access to early screening and care will be essential in reducing HNC

disparities. Investing in healthcare infrastructure, including training healthcare professionals and improving diagnostic facilities, is essential to address the disparities in HNC burden across different socioeconomic regions.

## Age-Related trends and the complex disease burden

This study also emphasizes age-related trends in HNC, with the incidence rising across most age groups. Nevertheless, a decrease in incidence among older populations indicates underdiagnosis or competing health issues that mask HNC symptoms [42,43]. The decreasing mortality and DALYs among younger populations suggest that advancements in early detection and treatment are especially beneficial for these groups. However, the risks associated with aging, including multiple comorbidities and heightened surgical risks, are significant factors which can influence HNC outcomes [34].

Further research is vital for better comprehending the biological and environmental factors contributing to the age-specific trends found in our analysis. The relationship between accelerated epigenetic aging and adverse outcomes in HNC highlights the need for personalized treatment approaches that address the unique challenges faced by elderly patients [44]. Raising awareness about the risk factors, symptoms, and prevention strategies for head and neck cancers can empower individuals to take proactive measures in reducing their risk and seeking timely medical attention.

## Study limitations and future directions

Although the obtained findings provide valuable insights into the global HNC burden, this study still had the following limitations. The GBD dataset, while comprehensive, is subject to variability in reporting accuracy and completeness across different regions [45]. Discrepancies in diagnostic and reporting practices may influence data comparability and introduce bias in this analysis. For instance, underreporting in low- and middle-SDI regions, where healthcare infrastructure may be less developed, could lead to underestimating the true disease burden in these area. This could skew global estimates and potentially mask critical trends, particularly in regions where HNC rates are expected to rise. The summary-level data applied in this study restricts the capability of conducting detailed mechanistic analyses, including those involving genetic or environmental factors. To mitigate these issues in future research, increasing the granularity of the data by incorporating patient-level information would allow for more refined analyses. Enhancing global health surveillance systems, particularly in under-resourced regions, could improve the accuracy and completeness of the data, thereby reducing bias. Collaborations with local health authorities to standardize diagnostic criteria and reporting practices could also help address regional disparities in data quality and comparability.

Future research should concentrate on incorporating individual-level data and exploring the complex interplay of genetic predispositions, environmental pollution, and lifestyle changes in shaping the global HNC burden. Key research questions could include: What specific genetic mutations or variants contribute to the higher incidence of HNC in certain populations? How do environmental pollutants, such as air and water contamination, interact with genetic predispositions to influence HNC outcomes? How do lifestyle factors, including smoking and alcohol consumption, modify these genetic and environmental risks? Expanding research into underrepresented populations and regions will be vital for refining our understanding of HNC dynamics and enhancing the precision of disease burden predictions. Moreover, fostering collaborations with epidemiologists, geneticists, and environmental scientists would provide an integrative approach, allowing for the development of more holistic models

considering both biological and external factors. Additionally, these interdisciplinary efforts could also lead to the identification of at-risk populations and the formulation of targeted public health interventions.

## Conclusions

To conclude, this study offers a detailed analysis of the global burden of head and neck cancer (HNC) using the GBD 2021 dataset, covering the period from 1980 to 2021, and introduces the HNCP-T model, a novel deep learning-based approach for the prediction of future trends from 2022 to 2030. Our model offers novel insights into the evolving patterns of HNC, especially highlighting significant regional and gender-based disparities. The projected rise in incidence is most pronounced in low- and middle-SDI regions and among women, driven by a combination of enhanced diagnostic capabilities and socio-economic factors. The superior performance of the transformer-based model in our comparative experiments highlights its effectiveness in capturing intricate patterns and dependencies, making it a valuable tool for future disease burden forecasting and public health planning. Through confirming our approach with real-world data, we exhibit the robustness and reliability of using deep learning models to forecast global health trends. This approach not only improves upon traditional statistical methods but also provides a powerful tool for more precise, data-driven public health strategies. The insights from the present study have the potential to inform more effective interventions, guide policy development, as well as better equip healthcare systems to address the growing burden of HNC globally. Our findings underscore the importance of continuing to refine predictive models to deal with global health challenges, particularly in regions and populations with increasing disease burdens. Moreover, the application of such models could cause more targeted and proactive public health responses in the future. However, to enhance the practical application of these findings, specific policy recommendations at local, national, and global levels should be explored. For instance, healthcare systems could adopt adaptive frameworks, integrating predictive models like HNCP-T, to forecast trends and allocate resources more effectively, focusing on high-risk regions and populations.

## Acknowledgments

We would like to express our sincere gratitude to the Global Burden of Disease (GBD) collaborators for providing the comprehensive dataset used in this study. We are particularly thankful to our colleagues at the Department of Otolaryngology, Second Affiliated Hospital, Anhui Medical University, and the School of Artificial Intelligence and Data Science, University of Science and Technology of China, for their valuable input and discussions that significantly contributed to this research.

## Author contributions

**Conceptualization:** Qiongyuan Hu, Shuai Lv, Xinyu Wang, Peng Pan.

**Data curation:** Qiongyuan Hu, Shuai Lv.

**Formal analysis:** Qiongyuan Hu, Shuai Lv.

**Funding acquisition:** Jinyu Mei.

**Investigation:** Qiongyuan Hu, Shuai Lv.

**Methodology:** Qiongyuan Hu, Shuai Lv.

**Project administration:** Qiongyuan Hu, Shuai Lv.

**Resources:** Qiongyuan Hu, Shuai Lv.

**Software:** Qiongyuan Hu, Shuai Lv.

**Supervision:** Qiongyuan Hu, Shuai Lv.

**Validation:** Qiongyuan Hu, Shuai Lv.

**Visualization:** Qiongyuan Hu, Shuai Lv.

**Writing – original draft:** Qiongyuan Hu, Shuai Lv, Wei Gong.

**Writing – review & editing:** Qiongyuan Hu, Shuai Lv, Xinyu Wang, Peng Pan, Jinyu Mei.

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
