## [Decision Letter · Decision Letter 0]

7 Jan 2025

PONE-D-24-49402Global Burden and Future Trends of Head and Neck Cancer: A Deep Learning-Based Analysis (1980 - 2030)PLOS ONE

Dear Dr.  Hu,

Thank you for submitting your manuscript to PLOS ONE. After careful consideration, we feel that it has merit but does not fully meet PLOS ONE’s publication criteria as it currently stands. Therefore, we invite you to submit a revised version of the manuscript that addresses the points raised during the review process.

We look forward to receiving your revised manuscript.

Kind regards,

Enes Erul, MD

Academic Editor

PLOS ONE

“This work was supported in part by the Anhui Provincial Science and Technology Department under Grant 2022AH050662, and in part by the Anhui Provincial Postgraduate Education Quality Engineering Project under Grant 2022zyxwjxalk060, and in part by the Research Fund of Anhui Institute of translational medicine under Grant 2022zhyx-C42, and  in part by the National Natural Science Foundation Incubation Program of The Second Affiliated Hospital of Anhui Medical University under Grant 2021GMFY04.”

Reviewers' comments:

Reviewer's Responses to Questions

**Comments to the Author**

1. Is the manuscript technically sound, and do the data support the conclusions?

Reviewer #1: Yes

Reviewer #2: Yes

2. Has the statistical analysis been performed appropriately and rigorously? 

Reviewer #1: Yes

Reviewer #2: N/A

3. Have the authors made all data underlying the findings in their manuscript fully available?

Reviewer #1: Yes

Reviewer #2: Yes

4. Is the manuscript presented in an intelligible fashion and written in standard English?

Reviewer #1: Yes

Reviewer #2: Yes

5. Review Comments to the Author

Reviewer #1: Thank you for submitting this interesting article examining the Global Burden and Future Trends of Head and Neck Cancer: A Deep Learning-Based Analysis (1980 - 2030).

The present study is a multicentric, retrospective study including data from 204 centers for head and neck cancer. Analyses focus on survival-standardized intra-person mortality and disability-adjusted life years (DALYs) for head and neck cancer. A Transformer-based model, HNCP-T, was used to predict trends up to 2030.

The study is well-presented and well-studied but needs a few minor corrections.

1. In the study, while analyzing the global changes in head and neck cancers, the use of the Transformer-based data model instead of LSTMs and RNNs is an appropriate and effective analytical method. The advantages of the Transformer-based model should be emphasized more clearly in the text.

2. When using predictor variables, Plasma Glucose Levels, High Body Mass Index (BMI), and Socio-Demographic Index (SDI) were included. However, given the increasing global burden of HPV and its significant association with head and neck cancers, HPV-related data should also be incorporated into the data packages.

3. Smoking and alcohol consumption undeniably influence the incidence of head and neck cancers. Figure 5 should include data on the impact of smoking and alcohol use, particularly concerning their correlation with socioeconomic levels. If this information is not available in the existing data, this point should be emphasized more clearly in the data analysis or discussion section.

4. Concrete recommendations derived from the study's findings, which can be utilized in public health policies, should be incorporated into the discussion section.

5.The term "the prediction model is consisted of" (Line 190) should be corrected to "the prediction model consists of.

6. The term "plays a important role in" on line 197 should be corrected to "plays an important role in.

7. The expression "gender-specific trends in HNC burden" (Line 374) should be revised to "gender-based trends," and the corresponding sections should be adjusted accordingly.

8. The term "By contrast" (Line 405) should be corrected to "In contrast."

9. The term "a decrease in incidence among older populations indicate" in line 429 should be corrected to "a decrease in incidence among older populations indicates."

10. The term "risks which are associated with aging" in line 433 should be corrected to "risks associated with aging."

11. The phrase "the need for personalized treatment approaches considering the unique challenges confronted by elderly patients" in line 437 should be replaced with "the need for personalized treatment approaches that address the unique challenges faced by elderly patients."

12. The term "where healthcare infrastructure may be less developed, could lead to underestimations of true disease burden in these areas" in lines 444-445 should be revised to "where healthcare infrastructure may be less developed, could lead to underestimating the true disease burden in these areas."

13. The phrase "especially highlighting significant regional and gender-specific disparities" in lines 477-478 is recommended to be revised to "especially highlighting significant regional and gender-based disparities."

Reviewer #2: In order to perform forecasting, the authors rely on the popular neural network based transformer architecture. As they correctly state in the manuscript, this architecture has many advantages compared to prior architectures such as RNNs. Moreover, since this is a deep learning based approach, it is advantageous in terms of modeling non-linear complex relationships. However, even though these are discussed verbally, they are not supported with results. More specifically, I believe that in addition to providing results for the proposed transformer architecture, the authors should provide results for alternative strategies. For instance, one alternative can be a classical statistical model, such as linear regression, and the other can be an alternative deep sequence learning based approach, such as LSTM. I think that this additional experiments are especially critical given that the data at hand is a time-series signal and categorical data. This is because, as opposed to natural language or vision tasks where transformers are absolute winners, in time series or tabular data, even classical methods are known to be able to outperform deep learning strategies frequently, especially when there are missing data or data is inherently noisy.

2) The idea for splitting the data based on the year, instead of random shuffling is very logical.

3) Could authors elaborate more on how the results on test set had guided them for better training?

6. PLOS authors have the option to publish the peer review history of their article (what does this mean?). If published, this will include your full peer review and any attached files.

Reviewer #1: No

Reviewer #2: No

---

## [Author Response · Author response to Decision Letter 1]

1 Feb 2025

Dear Professor Enes Erul,

Thank you very much for your e-mail dated on Jan. 08, 2025, informing us the editorial decision on our manuscript (PONE-D-24-49402). We would like to express our gratitude to you and the editor, as well as the anonymous reviewers for the time and effort spent in processing and

improving our paper. The constructive comments from you and the reviewers are very helpful for the improvement of our paper. This is to confirm that the paper has been duly revised in accordance with the comments made by you and the reviewers, which we would like to resubmit as a possible publication in the PLOS ONE . Attached files are the detailed responses to you and the reviewers, and the duly revised manuscript. Once again, we sincerely thank you for the time and effort you have spent and are going to spend in processing our paper. We look forward to hearing from you regarding its disposition.

Responses to Editor

Manuscript Code: PONE-D-24-49402

“Global Burden and Future Trends of Head and Neck Cancer: A Deep Learning-Based Analysis (1980 - 2030)”

We would like to express our sincere appreciation to you for your constructive comments and sug-gestions, and the time and effort you spent in helping us to improve the quality and presentation of the paper.

1. Journal requirements: Please ensure that your manuscript meets PLOS ONE’s style requirements, including those for file naming.

Response:Thank you for your valuable comment. We have now followed the Plos One template according to the submission guidelines and PLOS ONE’s style requirements, including those for file naming.

2. Journal requirements: Please note that PLOS ONE has specific guidelines on code sharing for submissions in which author-generated code underpins the findings in the manuscript. In these cases, all author-generated code must be made available without restrictions upon publication of the work. Please review our guidelines and ensure that your code is shared in a way that follows best practice and facilitates reproducibility and

reuse.

Response:Thank you for your valuable comment. We have reviewed the guidelines and ensure that our code is shared in the attachment called “Code. zip”.

3. Journal requirements: Thank you for stating the following financial disclosure :“This work was supported in part by the Anhui Provincial Science and Technology Department under Grant 2022AH050662, and in part by the Anhui Provincial Postgraduate Education Quality Engineering Project under Grant 2022zyxwjxalk060, and in part by the Research Fund of Anhui Institute of translational medicine under Grant 2022zhyx-C42, and in part by the National Natural Science Foundation Incubation Program of The Second Affiliated Hospital of Anhui Medical University under Grant 2021GMFY04.” Please state what role the funders took in the study. If the funders had no role, please state: ”The funders had no role in study design, data collection and analysis, decision to publish, or preparation of the manuscript.” If this statement is not correct you must amend it as needed. Please include this amended Role of Funder statement in your cover letter; we will change the online submission form on your behalf. Response:Thank you for your valuable comment. we have corrected the financial disclo-sure. Kindly include the following text on the online submission form: “This work was supported in study design by the Anhui Provincial Science and Tech-nology Department under Grant 2022AH050662, and in data collection and analysis by the Anhui Provincial Postgraduate Education Quality Engineering Project under Grant

2022zyxwjxalk060, and in decision to publish by the Research Fund of Anhui Institute of translational medicine under Grant 2022zhyx-C42, and in preparation of the manuscript by the National Natural Science Foundation Incubation Program of The Second Affiliated Hospital of Anhui Medical University under Grant 2021GMFY04.”

4. Journal requirements: Please review your reference list to ensure that it is complete and correct. If you have cited papers that have been retracted, please include the rationale for doing so in the manuscript text, or remove these references and replace them with relevant

current references. Any changes to the reference list should be mentioned in the rebuttal letter that accompanies your revised manuscript. If you need to cite a retracted article, indicate the article’s retracted status in the References list and also include a citation and full reference for the retraction notice.

Response:Thank you for your valuable comment. We have added six new references . Based on the comments of the reviewers, we realized that we needed to further elaborate on the superiority of the transformer-based model, so we further conducted a comparative experiment, and the model used for comparison appeared in two articles, so we cited two articles newly (see Section III, Page 17). At the same time, We have added more concrete recommendations in the discussion section (see Section V, Page 18 and 20), so we cited four articles newly.

[23] Rauniyar SK, Hashizume M, Yoneoka D, Nomura S. Projection of morbidity and mortality due to breast cancer between 2020 and 2050 across 42 low-and middle-income countries. Heliyon. 2023; 9(6). 644

[24] Zhao Q, Chen M, Fu L, Yang Y, Zhan Y. Assessing and projecting the global burden of thyroid cancer, 1990–2030: Analysis of the Global Burden of Disease Study. Journal of Global Health. 2024; 14. 647

[25] Manthey J, Shield KD, Rylett M, Hasan OSM, Probst C, Rehm J. Global alcohol exposure between 1990 and 2017 and forecasts until 2030: a modelling study. The Lancet. 2019; 393(10190): 2493–2502.

[26] Jinyi W, Zhang Y, Wang K, Peng P. Global, regional, and national mortality of tuberculosis attributable to alcohol and tobacco from 1990 to 2019: A modelling study based on the Global Burden of Disease study 2019. Journal of Global Health. 2024;14.

[27] Moscowchi A, Moradian-Lotfi S, Koohi H, Sarrafan Sadeghi T. Levels of smoking and outcome measures of root coverage procedures: a systematic review and meta-analysis. Oral and maxillofacial surgery. 2024; 28(2): 485–497.

[28] Khan Minhas AM, Sedhom R, Jean ED, Shapiro MD, Panza JA, Alam M, et al. Global burden of cardiovascular disease attributable to smoking, 1990–2019: an analysis of the 2019 Global Burden of Disease Study. European Journal of Preventive Cardiology. 2024; 31(9): 1123–1131.

Responses to Reviewer #1

Manuscript Code: PONE-D-24-49402

“Global Burden and Future Trends of Head and Neck Cancer: A Deep Learning-Based Analysis (1980 - 2030)”

We would like to express our sincere appreciation to you for your constructive comments and sug-gestions, and the time and effort you spent in helping us to improve the quality and presentation of the paper.

1. Comment: In the study, while analyzing the global changes in head and neck cancers, the use of the Transformer-based data model instead of LSTMs and RNNs is an appropriate and effective analytical method. The advantages of the Transformer-based model should be emphasized more clearly in the text.

Response: Thank you for your valuable comment. We further conducted comparative experiments to clear the advantages of the transformer-based model. The LSTM model is used to project the global burden of breast cancer between 2020 and 2050 in [23], while a linear regression model is employed to project the global burden of thyroid cancer between 2020 and 2030 in [24]. Therefore, we incorporate these models as benchmarks to evaluate the effectiveness of our proposed approach (see Section III, Page 17). At the same time, we have also added more discussion about the advantages of the transformer-based model and the reasons why this model is used in this work in Introduction (see Section I, Page 2), Materials and methods (see Section II, Page 5), Discussion (see Section V, Page 18), and Conclusions (see Section VI, Page 21). These changes ensure that the benefits of the transformer-based model are more clearly communicated and supported by empirical evidence, strengthening the justification for its use in this study.

[23] Rauniyar SK, Hashizume M, Yoneoka D, Nomura S. Projection of morbidity and mortality due to breast cancer between 2020 and 2050 across 42 low-and middle-income countries. Heliyon. 2023; 9(6). 644

[24] Zhao Q, Chen M, Fu L, Yang Y, Zhan Y. Assessing and projecting the global burden of thyroid cancer, 1990–2030: Analysis of the Global Burden of Disease Study. Journal of Global Health. 2024; 14. 647

2. Comment: When using predictor variables, Plasma Glucose Levels, High Body Mass Index (BMI), and Socio-Demographic Index (SDI) were included. However, given the increasing global burden of HPV and its significant association with head and neck cancers, HPV-related data should also be incorporated into the data packages.

Response: Thanks for the comment. We sincerely appreciate the reviewer for highlighting the importance of HPV-related data in the analysis of head and neck cancers (HNC). We acknowledge that Human Papillomavirus (HPV) is a significant risk factor contributing to the incidence of HNC, and its increasing global burden warrants attention. However, we reconfirmed the GBD2021 dataset and found that it does not contain HPV-related data, which is why we did not include it in our data packages. To address this important factor, we have expanded the Discussion section to emphasize the role of HPV in the global burden of HNC (see Section V, Page 18).

3. Comment: Smoking and alcohol consumption undeniably influence the incidence of head and neck cancers. Figure 5 should include data on the impact of smoking and alcohol use, particularly concerning their correlation with socioeconomic levels. If this information is

not available in the existing data, this point should be emphasized more clearly in the data analysis or discussion section.

Response: Thanks for the comment. We have expanded the Discussion section to em-phasize the role of smoking and alcohol consumption in the global burden of HNC (see Section V, Page 18).

4. Comment: Concrete recommendations derived from the study’s findings, which can be utilized in public health policies, should be incorporated into the discussion section.

Response: Thanks for the comment. We have added more concrete recommendations in the discussion section (see Section V, Page 18 and 20).

5. Comment: The term “the prediction model is consisted of ” (Line 190) should be corrected to “the prediction model consists of ”.

Response: We sincerely thank the reviewer for careful reading. As suggested by the reviewer, we have corrected the “the prediction model is consisted of” into “the prediction model consists of”.(see Page 6, line 189)

6. Comment: The term “plays a important role in” on line 197 should be corrected to “plays an important role in”.

Response: We sincerely thank the reviewer for careful reading. As suggested by the reviewer, we have corrected the “plays a important role in” into “plays an important role in”.(see Page 6, line 196)

7. Comment: The expression “gender-specific trends in HNC burden” (Line 374) should be revised to “gender-based trends”, and the corresponding sections should be adjusted accordingly.

Response: We sincerely thank the reviewer for careful reading. As suggested by the reviewer, we have corrected the “gender-specific trends in HNC burden” into “gender-based trends”, and the corresponding sections have adjusted accordingly.(see Page 19 and 21, line 432, 433, 454, 541)

8. Comment: The term “By contrast” (Line 405) should be corrected to “In contrast”.

Response: We sincerely thank the reviewer for careful reading. As suggested by the reviewer, we have corrected the “By contrast” into “In contrast”.(see Page 19, line 463)

9. Comment: The term “a decrease in incidence among older populations indicate” in line 429 should be corrected to “a decrease in incidence among older populations indicates”.

Response: We sincerely thank the reviewer for careful reading. As suggested by the reviewer, we have corrected the “a decrease in incidence among older populations indicate” into “a decrease in incidence among older populations indicates”.(see Page 20, line 491)

10. Comment: The term “risks which are associated with aging” in line 433 should be corrected to “risks associated with aging”.

Response: We sincerely thank the reviewer for careful reading. As suggested by the reviewer, we have corrected the “risks which are associated with aging” into “risks associated with aging”.(see Page 20, line 494)

11. Comment: The phrase “the need for personalized treatment approaches considering the unique challenges confronted by elderly patients” in line 437 should be replaced with “the need for personalized treatment approaches that address the unique challenges faced by elderly patients”.

Response: We sincerely thank the reviewer for careful reading. As suggested by the reviewer, we have corrected the “the need for personalized treatment approaches consid-ering the unique challenges confronted by elderly patients” into “the need for personalized treatment approaches that address the unique challenges faced by elderly patients”.(see Page 20, line 499)

12. Comment: The term “where healthcare infrastructure may be less developed, could lead to underestimations of true disease burden in these areas” in lines 444-445 should be revised to ”where healthcare infrastructure may be less developed, could lead to underestimating

the true disease burden in these area”.

Response: We sincerely thank the reviewer for careful reading. As suggested by the reviewer, we have corrected the “where healthcare infrastructure may be less developed, could lead to underestimations of true disease burden in these areas” into “where healthcare

infrastructure may be less developed, could lead to underestimating the true disease burden in these area”.(see Page 20, line 510)

13. Comment: The phrase “especially highlighting significant regional and gender-specific

disparities” in lines 477-478 is recommended to be revised to “especially highlighting significant regional and gender-based disparities”.

Response: We sincerely thank the reviewer for careful reading. As suggested by the reviewer, we have corrected the “especially highlighting significant regional and gender-specific disparities” into “especially highlighting significant regional and gender-based disparities”.(see Page 21, line 541) Once again, the authors would like to thank you for your precious comments and suggestions.

Without these comments and suggestions, the paper would not be improved to its present quality.

Responses to Reviewer #2

Manuscript Code: PONE-D-24-49402

“Global Burden and Future Trends of Head and Neck Cancer: A Deep Learning-Based Analysis (1980 - 2030)”

We would like to express our sincere appreciation to you for your constructive comments and suggestions, and the time and effects you spent in helping us to improve the quality and presen-tation of the paper.

1. Comment: In order to perform forecasting, the authors rely on the popular neural network based transformer architecture. As they correctly state in the manuscript, this architecture has many advantages compared to prior architectures such as RNNs. Moreover, since this is a deep learning based approach, it is advantageous in terms of modeling non-linear complex relationships. However, even though these are discussed verbally, they are not supported with results. More specifically, I believe that in addition to providing results for the proposed transformer architecture, the authors should provide results for alternative strategies. For instance, one alternative can be a classical statistical model, such as linear regression, and the other can be an alternative deep sequence learning based approach, such as LSTM. I think that this additional experiments are especially critical given that the data at hand is a time-series signal and categorical data. This is because, as opposed to natural

language or vision tasks where transformers are absolute winners, in time series or tabular data, even classical methods are known to be able to outperform deep learning strategies frequently, especially when there are missing data or data is inherently noisy.

Response: Thank you for your valuable comment.

---

## [Decision Letter · Decision Letter 1]

17 Feb 2025

Global Burden and Future Trends of Head and Neck Cancer: A Deep Learning-Based Analysis (1980 - 2030)

PONE-D-24-49402R1

Dear Dr. Qiongyuan Hu,

We’re pleased to inform you that your manuscript has been judged scientifically suitable for publication and will be formally accepted for publication once it meets all outstanding technical requirements.

Kind regards,

Enes Erul, MD

Academic Editor

PLOS ONE

Reviewers' comments:

Reviewer's Responses to Questions

**Comments to the Author**

1. If the authors have adequately addressed your comments raised in a previous round of review and you feel that this manuscript is now acceptable for publication, you may indicate that here to bypass the “Comments to the Author” section, enter your conflict of interest statement in the “Confidential to Editor” section, and submit your "Accept" recommendation.

Reviewer #1: All comments have been addressed

Reviewer #2: All comments have been addressed

2. Is the manuscript technically sound, and do the data support the conclusions?

Reviewer #1: Yes

Reviewer #2: Yes

3. Has the statistical analysis been performed appropriately and rigorously? 

Reviewer #1: Yes

Reviewer #2: Yes

4. Have the authors made all data underlying the findings in their manuscript fully available?

Reviewer #1: Yes

Reviewer #2: Yes

5. Is the manuscript presented in an intelligible fashion and written in standard English?

Reviewer #1: Yes

Reviewer #2: Yes

6. Review Comments to the Author

Reviewer #1: Thank you for submitting this interesting article examining the Global Burden and Future Trends of Head and Neck Cancer: A Deep Learning-Based Analysis (1980 - 2030).

Your findings provide important insights into regional and demographic disparities in HNC burden and emphasize the need for data-driven decision-making in healthcare resource allocation and preventive strategies.

The use of transformer-based deep learning models for forecasting future trends (2022-2030) has been particularly noted as a valuable advancement in epidemiological research.

The changes suggested in the previous reviews have been implemented, and the manuscript is now deemed suitable for publication in the journal.

Reviewer #2: I sincerely appreciate your time and effort in reviewing our manuscript. Thank you for your valuable feedback and thoughtful comments. Your insights have been truly appreciated.

7. PLOS authors have the option to publish the peer review history of their article (what does this mean?). If published, this will include your full peer review and any attached files.

Reviewer #1: No

Reviewer #2: No

---

## [Editor Report · Acceptance letter]

PONE-D-24-49402R1

PLOS ONE

Dear Dr. Hu,

I'm pleased to inform you that your manuscript has been deemed suitable for publication in PLOS ONE. Congratulations! Your manuscript is now being handed over to our production team.

Kind regards,

on behalf of

Dr. Enes Erul

Academic Editor

PLOS ONE